# Impact of aging and exercise on skeletal muscle mitochondrial capacity, energy metabolism, and physical function

L. Grevendonk[1,2], N. J. Connell[1,2], C. McCrum [1], C. E. Fealy[1,2], L. Bilet[1,2], Y. M. H. Bruls[3], J. Mevenkamp[3], V. B. Schrauwen-Hinderling [1,3], J. A. Jörgensen[1], E. Moonen-Kornips[1], G. Schaart [1], B. Havekes[1,4], J. de Vogel-van den Bosch [5], M. C. E. Bragt[6], K. Meijer [1], P. Schrauwen [1,2] & J. Hoeks [1,2✉]

The relationship between the age-associated decline in mitochondrial function and its effect on skeletal muscle physiology and function remain unclear. In the current study, we examined to what extent physical activity contributes to the decline in mitochondrial function and muscle health during aging and compared mitochondrial function in young and older adults, with similar habitual physical activity levels. We also studied exercise-trained older adults and physically impaired older adults. Aging was associated with a decline in mitochondrial capacity, exercise capacity and efficiency, gait stability, muscle function, and insulin sensitivity, even when maintaining an adequate daily physical activity level. Our data also suggest that a further increase in physical activity level, achieved through regular exercise training, can largely negate the effects of aging. Finally, mitochondrial capacity correlated with exercise efficiency and insulin sensitivity. Together, our data support a link between mitochondrial function and age-associated deterioration of skeletal muscle.

[1] Department of Nutrition and Movement Sciences, NUTRIM School of Nutrition and Translational Research in Metabolism, Maastricht University, Maastricht, The Netherlands. [2] TI Food and Nutrition, Wageningen, The Netherlands. [3] Department of Radiology and Nuclear Medicine, NUTRIM School of Nutrition and Translational Research in Metabolism, Maastricht University Medical Center+, Maastricht, The Netherlands. [4] Department of Internal Medicine, Division of Endocrinology, NUTRIM School of Nutrition and Translational Research in Metabolism, Maastricht University Medical Center+, Maastricht, The Netherlands. [5] Danone Nutricia Research, Utrecht, The Netherlands. [6] Friesland-Campina, Amersfoort, The Netherlands. ✉email: j.hoeks@maastrichtuniversity.nl

A large portion of the world's population is of middle to older age, and in high-income countries, the proportion of people over the age of 60 is growing faster than any other age group[1]. Since aging is associated with a variety of comorbidities resulting in rapidly increasing health care costs, this demographic change is one of the major societal challenges of the current decade[2].

One of the distinctive features of aging is the progressive loss of muscle mass and physical function, collectively known as sarcopenia[3]. The loss of skeletal muscle mass and tissue function has been related to mobility impairments[4], such as difficulties walking short distances or standing up from a chair[5,6], an increased risk of falls[7,8], physical frailty[9], and metabolic impairments[10], ultimately leading to a loss of physical independence and increased care need[11]. Improving balance and mobility are important factors for public health in older adults in order to reduce fall-related consequences such as fractures, further functional decline, immobility, and death[12].

In parallel with the progressive loss of muscle function, mitochondrial respiratory activity in human skeletal muscle has been shown to decrease with advancing age in healthy men and women[13,14]. Furthermore, protein levels of the mitochondrial master regulator peroxisome proliferator-activated receptor gamma co-activator 1α (PGC-1α) were found to correlate with walking speed in healthy older adults[15]. Some preclinical studies indeed suggest that the reduction in muscle mitochondrial function may underlie the decline in muscle health during aging. Thus, an accelerated manifestation of sarcopenia is seen in transgenic mice lacking the antioxidant enzyme superoxide dismutase 1 (SOD1), which are characterized by both a diminished mitochondrial bioenergetic function and an induction of mitochondrial-mediated apoptosis[16]. In addition, in rats, age-related increases in mitochondrial abnormalities have been associated with muscle fiber splitting and atrophy[17]. Therefore, it is tempting to speculate that augmenting mitochondrial function could be a potential strategy to counteract aging-associated decline in physical function.

Although some human studies have addressed age-related alterations in muscle mitochondrial function in relation to the decline in skeletal muscle function[5,18–21], the available data in humans is scarce and the few available studies often focus on either the decline in muscle function or concentrate primarily on the mitochondrial alterations. Additionally, the age-associated decline in mitochondrial function is not completely attributable to aging per se and may also be explained, in part, by an age-related decline in physical activity (PA)[19,20]. Decreased PA can adversely affect mitochondrial capacity[22], while exercise training stimulates mitochondrial biogenesis through increases in the master regulator PGC-1α[23]. Older adults are more inactive with advancing age which in turn decreases functional fitness[24]. Therefore, it remains unclear to what extent chronological age and physical inactivity contribute to the decline in mitochondrial function and muscle health during aging.

To delineate these relationships, we conducted a cross-sectional study with detailed phenotyping in groups of young versus older human participants, with a range in oxidative capacity and physical function. The first aim of the study was to assess if mitochondrial function is reduced in older compared to young participants with a similar level of habitual PA, and to examine how mitochondrial function relates to muscle function. To this end, we performed a wide array of assessments of skeletal muscle function (including strength, volume, insulin sensitivity, gait stability and adaptability, exercise capacity, and exercise efficiency) as well as in and ex vivo measurements to characterize mitochondrial capacity. The second aim of the study was to investigate the potential of regular exercise training in older adults for the maintenance of mitochondrial function, and if this is associated with maintained muscle health. Using the methods described above, the older adults with normal PA levels were also compared to older adults with a high PA level (trained individuals) and older adults with low physical function (physically impaired individuals, low PA). We also performed correlation analyses between skeletal muscle mitochondrial function, physical function, and muscle health during aging.

## Results

**Mitochondrial function and muscle health decrease with age.** To assess the effect of age on mitochondrial function and muscle health, we compared young (Y) versus older (O) individuals with comparable levels of PA. Table 1 summarizes the main participant characteristics. Average age was 24 (3) years for the young individuals (Y) and 71 (4) years for the older participants ($p < 0.001$). Sex distribution was identical between the two groups (8 male and 9 female participants in both groups, $p > 0.999$). Older individuals had an increased BMI ($p = 0.007$) and an increased body fat percentage ($p = 0.007$) as compared to young individuals. Fat-free mass (FFM) was similar in young adults and old individuals and ($p = 0.471$). Both age groups performed a comparable amount of steps per day ($p = 0.146$) and showed no significant difference in time spent on high PA level or low PA level ($p = 0.496$ and $p = 0.737$, respectively) indicating comparable levels of PA in daily life between young and older participants. Furthermore, no significant differences were observed in the balance test, the 4-meter walk test, and chair-stand test, assessed during screening as part of the short physical performance battery (SPPB, $p > 0.05$).

**Lower muscle strength, volume, and endurance in older adults.** During the 6-minute walk test (6MWT), older adults covered ~9% less distance, in comparison with the young participants ($p = 0.032$, Fig. 1a). Older adults also had a lower cardiorespiratory fitness, as indicated by a ~26% lower maximal aerobic capacity (VO$_2$max), in comparison with the young individuals

**Table 1 Participant body composition characteristics, physical activity, and physical function.**

|  | Young, normal physical activity (Y) | Older adults, normal physical activity (O) |
|---|---|---|
| N | 17 | 17 |
| Gender F/M[a] | 8/9 | 8/9 |
| Age (years) | 24 (3)[b] | 71 (4) |
| BMI (kg m$^{-2}$) | 22.7 (2.9)[b] | 25.8 (3.3) |
| Body weight (kg) | 69.0 (10.2) | 74.0 (11.9) |
| Fat mass (%) | 24.4 (8.2)[b] | 33.1 (9.2) |
| Fat mass (kg) | 16.8 (6.1)[b] | 24.5 (7.9) |
| Fat-free mass (kg) | 52.2 (9.9) | 49.5 (10.8) |
| Steps/day | 10,075 (3160) | 9983 (2781) |
| HPA time (% of wake time) | 2.6 (1.9) | 2.2 (1.3) |
| LPA time (% of wake time) | 11.4 (4.4) | 11.4 (2.4) |
| SPPB 4 m walk speed (m s$^{-1}$) | 1.1 (0.2) | 1.1 (0.2) |
| SPPB Chair-stand test (s) | 9.1 (2.6) | 10.1 (1.4) |

Data are presented as mean (SD). Body composition and physical activity could not be measured in one participant of the Y group due to the implications of the SARS-CoV-19 crisis.
[a]Sex distribution across groups was tested by $\chi^2$ test ($p > 0.999$).
[b]Indicates significant difference between 2 groups ($p < 0.05$ two-sided, independent samples t-test).
*BMI* body mass index, *HPA* high-intensity physical activity, *LPA* low-intensity physical activity, *SPPB* short physical performance battery.

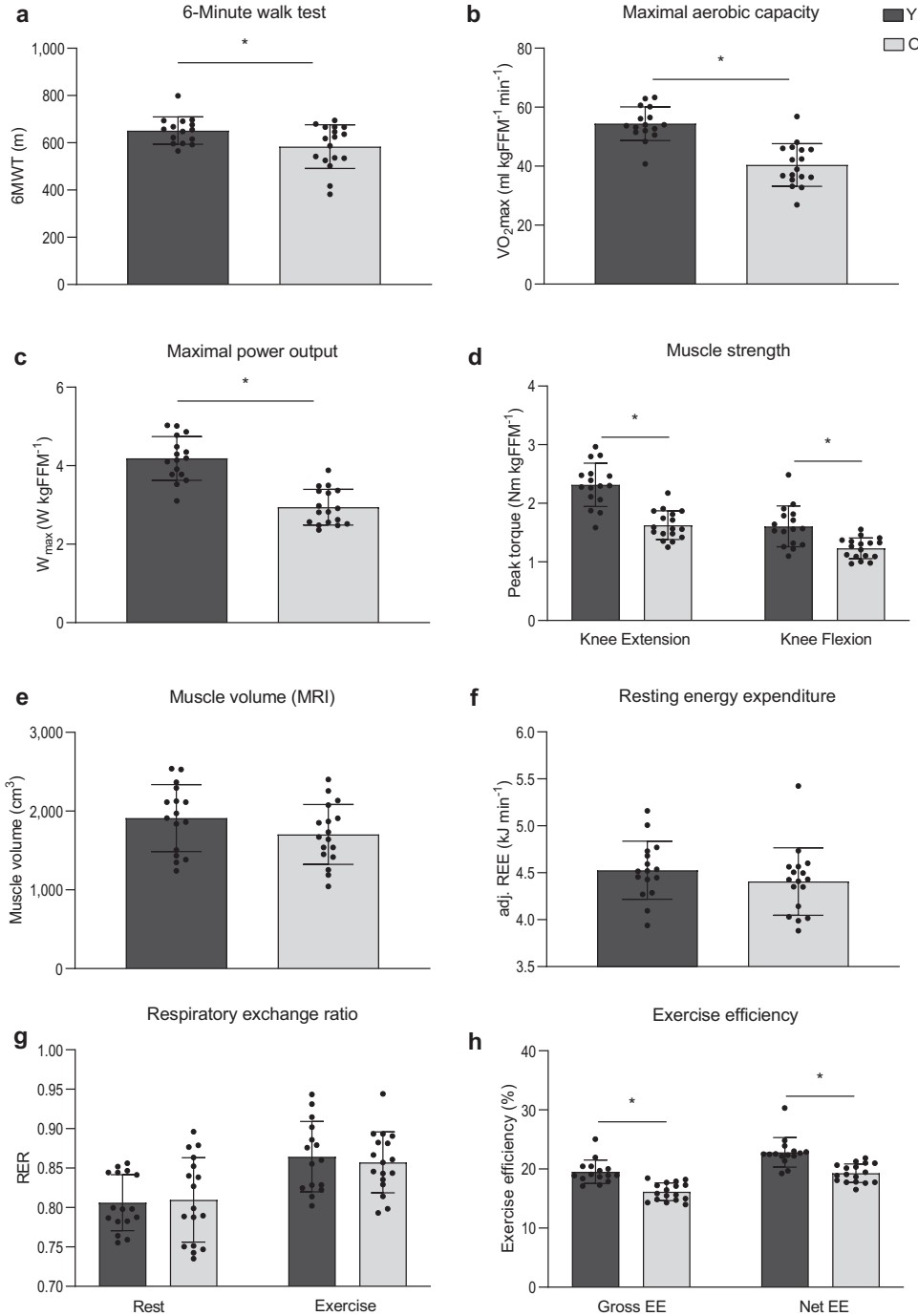

**Fig. 1 Effect of aging on muscle health and exercise efficiency. a** Walking distance during the 6MWT performed on the Caren-system (Y, $n = 14$; O, $n = 17$). **b** Maximum rate of oxygen consumption measured during graded cycling exercise corrected for FFM (Y, $n = 16$; O, $n = 17$). **c** Maximal power output measured during graded cycling test (Y, $n = 16$; E, $n = 17$). **d** Muscle strength expressed as the extension and flexion peak torque during an isokinetic protocol on the Biodex system and corrected for fat-free mass (Y, $n = 16$; O, $n = 17$). **e** Upper leg muscle volume measured by MRI (Y, $n = 17$; O, $n = 17$). **f** Resting energy expenditure adjusted for fat-free mass (Y, $n = 15$; O, $n = 17$). **g** Respiratory exchange ratio measured before, in resting conditions, and during submaximal exercise (Y, $n = 15$; O, $n = 17$). **h** Exercise efficiency measured during the submaximal cycle test and expressed as gross efficiency and net efficiency (Y, $n = 15$; O, $n = 17$). Dark gray bars represent the young individuals (Y), light gray bars represent the older individuals (O). FFM could not be measured in one Y due to the implications of the SARS-CoV-19 outbreak. For the same reason, 6MWT data and muscle volume data are missing in the same Y participant. Another participant from Y did not perform the 6MWT due to scheduling issues. The reported 6MWT distance from one Y participant was invalid and therefore excluded for analysis. One Y participant failed to complete the submaximal cycle test due to exhaustion. In another Y participant, the test could not be performed due to the implications of the SARS-CoV-19 outbreak. Values are presented as mean ± SD (with individual data points), Asterisk denotes significant differences between the two groups ($p < 0.05$, two-sided, independent samples $t$-test). 6MWT 6-min walk test, VO₂max maximal oxygen flow, Nm newton meters, FFM fat-free mass, RER respiratory exchange ratio, REE resting energy expenditure, EE exercise efficiency.

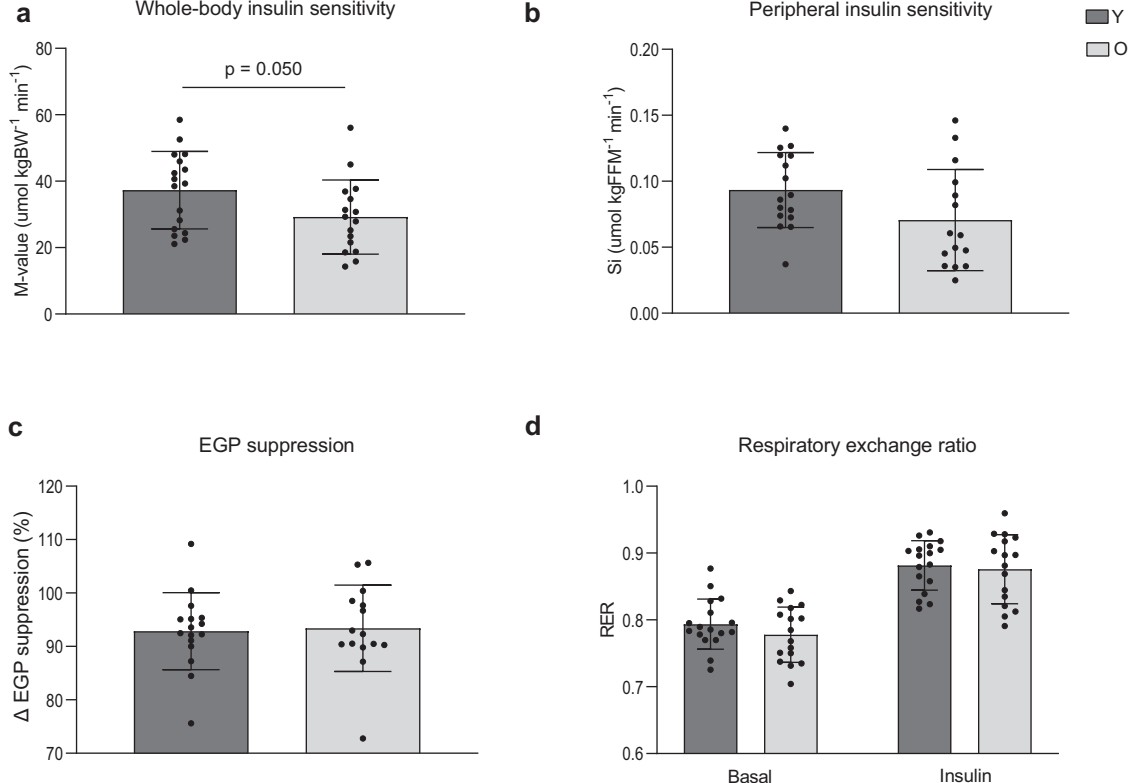

**Fig. 2 Effect of aging on insulin sensitivity and substrate selection. a** Whole-body insulin sensitivity based on the *M*-value measured during a hyperinsulinemic-euglycemic clamp. **b** Insulin-stimulated glucose uptake, corrected for plasma insulin and glucose levels ($S_i$) and expressed per kg FFM. **c** EGP suppression, calculated as the percentage insulin-suppressed EGP from the basal EGP **d** Respiratory exchange ratio measured before, in resting conditions (Basal), and during insulin stimulation (Insulin). Dark gray bars represent the young individuals (Y, $n = 17$), light gray bars represent older individuals (O, $n = 15$). One participant from O was excluded for analysis due to a violation of the protocol instructions. For one O participant, tracer data could not be analyzed. Values are presented as mean ± SD (with individual data points), $p = 0.05$ denotes a significant difference between the two groups (two-sided, independent samples *t*-test). *M*-value mean glucose infusion rate, BW body weight, $S_i$ insulin-stimulated glucose disposal, FFM fat-free mass, EGP endogenous glucose production.

($p < 0.001$, Fig. 1b). Likewise, maximal power output was lower in older individuals as compared to young ($p < 0.001$, Fig. 1c). In addition, a lower isokinetic muscle strength was observed in older adults as compared to young ($p < 0.001$, Fig. 1d). In line with the similar FFM, muscle volume, measured in the upper leg by MRI, was comparable between young and older individuals ($p = 0.151$, Fig. 1e).

**Reduced exercise efficiency in older adults**. Besides maximal exercise capacity, we also assessed exercise efficiency and substrate utilization during a submaximal exercise test. Resting energy expenditure was found to be comparable between young and old ($p = 0.275$, Fig. 1f) and no differences were observed in respiratory exchange ratio (RER) between old and young, both when measured during the submaximal exercise ($p = 0.624$, Fig. 1g) and resting conditions ($p = 0.852$, Fig. 1g). Both gross ($p < 0.001$) and net ($p < 0.001$) exercise efficiency were found to be lower in the older individuals as compared to the young participants (Fig. 1h).

**Lower whole-body insulin sensitivity in older adults**. To further characterize skeletal muscle health, insulin sensitivity was determined during a hyperinsulinemic-euglycemic clamp. During the clamp, the *M*-value required to maintain euglycemia, was ∼22% lower in older in comparison with young individuals ($p = 0.050$, Fig. 2a), suggesting a lower whole-body insulin sensitivity in older adults. Plasma insulin levels during the clamp were different

between young and older adults (Y: 75.61 (9.83) mU l⁻¹ vs. O: 88.29 (10.73) mU l⁻¹), $p = 0.001$). Peripheral insulin sensitivity, defined as the insulin-stimulated glucose uptake (Rd) corrected for differences in insulin and glucose levels ($S_i$), was also significantly lower in older vs. young individuals when expressed per kg body weight (O: 0.050 (0.028) μmol kgBW⁻¹ min⁻¹ vs. Y: 0.073 (0.024) μmol kgBW⁻¹ min⁻¹, $p = 0.014$). This difference was largely retained when $S_i$ was corrected for FFM, although it no longer reached statistical significance ($p = 0.069$, Fig. 2b). This may suggest that a small part of the difference in peripheral insulin sensitivity between young and old adults is related to lean mass. Insulin-mediated suppression of endogenous glucose production (%EGP suppression, $p = 0.850$, Fig. 2c) was not different between young and old. RER in basal conditions was also comparable between older and young individuals ($p = 0.174$, Fig. 2d). Likewise, upon insulin stimulation, RER was found to be equal between the older and young groups ($p = 0.740$, Fig. 2d). As a result, the change in RER upon insulin stimulation (known as metabolic flexibility) was also not different between older and young individuals (0.10 (0.05) vs. 0.09 (0.03), respectively, $p = 0.603$).

**Less gait stability but comparable adaptability in older adults**. To further determine physical function, walking stability was examined via step variability during unperturbed walking trials at multiple speeds on a treadmill and via stability of the body configuration (margin of stability; MoS;[25]) in response to

treadmill belt acceleration perturbations causing forward balance loss during walking (see Methods section for protocol details). Compared to young, older participants had a larger step length variability during unperturbed walking ($F_{(1,30)} = 7.077$, $p = 0.012$; significant pairwise comparisons at 1.2, 1.4, and 1.6 m s$^{-1}$; Supplementary Fig. 1), but no other significant differences were found. During the walking perturbation assessments, group comparisons for the entire perturbation trial revealed a significant difference in the margin of stability (MoS) during the first perturbation ($F_{(1,28)} = 7.7$, $p = 0.010$) but not the second and ninth perturbations, indicating that older adults initially perform poorer than young adults, but the ability of older adults to adapt gait in response to repeated perturbations is intact (Supplementary Fig. 2a). More detailed results regarding gait stability and stepping behavior can be found in the supplementary notes and accompanying figures.

**Lower ex vivo and in vivo mitochondrial function in older adults.** Skeletal muscle mitochondrial capacity was determined both ex vivo in permeabilized muscle fibers and in vivo via non-invasive assessment of PCr recovery rates using $^{31}$P-MR spectroscopy. In permeabilized muscle fibers, mitochondrial state 2 respiration (i.e., respiration in the presence of substrate alone) was similar between young and older adults on most substrate combinations, apart from malate + octanoyl-carnitine, which showed lower respiration rates in older individuals (MO, $p = 0.044$, Fig. 3a). ADP-stimulated (state 3) respiration, fueled by complex I-linked substrates (malate + glutamate, MG, $p = 0.032$, Fig. 3b), was lower in older participants whereas state 3 respiration upon a lipid substrate (MO, Fig. 3c) was not significantly different between young and old ($p > 0.05$). ADP-stimulated respiration upon parallel electron input to both Complex I and II was also lower in older compared to young individuals. Thus, state 3 respiration upon malate + octanoyl-carnitine + glutamate (MOG, $p = 0.014$, Fig. 3d), malate + octanoyl-carnitine + glutamate + succinate (MOGS, $p = 0.006$, Fig. 3d) and malate + glutamate + succinate (MGS, $p = 0.077$, Fig. 6d) was ~11–15% lower in the older adults as compared to the young individuals, although the latter did not reach statistical significance. Maximal FCCP-induced uncoupled respiration, reflecting the maximal capacity of the electron transport chain, was ~17% lower in older adults compared to young (State 3u, $p = 0.008$, Fig. 3e). State 4o respiration, reflecting mitochondrial proton leak, was similar between the young and older individuals ($p = 0.379$, Fig. 3f). The negligible increase in oxygen consumption upon cytochrome C (2.30 (3.24) vs. 2.27 (2.26) % in young vs. old, respectively) underscores the viability and quality of the muscle fibers and was similar in both groups ($p = 0.976$). Assessment of in vivo mitochondrial function revealed a ~16% lower PCr recovery rate constant in older adults as compared to young ($p = 0.003$, Fig. 3g), further confirming a decreased mitochondrial oxidative capacity in older individuals. The exercise-induced PCr depletion and end-exercise pH were similar between young and old ($p > 0.05$) and pH remained above 6.9, indicating no substantial exercise-induced acidification in either group. To complement the in vivo and ex vivo mitochondrial function measurements, mitochondrial content was estimated by mitochondrial oxidative phosphorylation (OXPHOS) protein expression. However, young and older adults displayed a similar expression of each of the five OXPHOS proteins ($p > 0.05$, Fig. 3h).

**Exercise training affects mitochondrial function and muscle health.** Next, we compared muscle function and mitochondrial capacity of the same older adults with normal PA (O, $n = 17$) to

trained older adults (TO, $n = 19$) as well as to older adults with a low physical function (IO, $n = 6$). Table 2 summarizes the main participant characteristics of these three older groups. Sex distribution was comparable across the groups ($p = 0.982$). Average age was 71 (4) years for the older participants with normal PA (O) and 71 (5) years for the physically impaired older adults (IO), whereas TO were slightly younger at 68 (2) years in comparison with O ($p = 0.023$). In accordance with their training status, TO displayed a lower BMI and a lower body fat mass percentage as compared to both O ($p = 0.046$ and $p = 0.050$, respectively) and IO ($p = 0.024$ and $p = 0.021$, respectively). FFM was comparable ($p > 0.05$) across the three groups. TO displayed a higher average step count per day as compared to O ($p = 0.050$) and IO ($p = 0.004$), further underscoring their active lifestyle. TO also spent a significantly higher proportion of their waking time at a high intensity level of PA compared to O ($p = 0.049$) and IO ($p = 0.005$). Proportion of the waking time spent at a low-intensity level of PA was similar between TO and O and between TO and IO ($p > 0.05$) and higher in O compared to IO ($p = 0.038$). As anticipated, IO performed significantly poorer on the chair-stand test as compared to O ($p = 0.015$) and TO ($p = 0.001$). Walking speed during the 4-meter walk test was also lower in IO when compared to TO ($p = 0.009$) but not to O ($p = 0.227$).

**Higher muscle strength, volume, and endurance in trained older adults.** During the 6MWT, IO covered ~22% less distance compared to the O group ($p = 0.013$, Fig. 4a) and ~29% less distance compared to the TO group ($p = 0.005$, Fig. 4a). Not surprisingly, the trained older individuals displayed the highest cardiorespiratory fitness levels, as exemplified by an average VO$_2$max which was ~1.2-fold and ~1.4-fold higher in comparison with O ($p = 0.003$) and IO ($p = 0.004$), respectively (Fig. 4b). Accordingly, maximal power output was higher in TO as compared to O and IO ($p < 0.001$ and $p < 0.001$ respectively, Fig. 4c). TO participants also showed the highest muscle strength, whereas muscle strength was lowest in the physically impaired older adults. Thus, isokinetic extension peak torque (Fig. 4d) was ~24% and ~27% lower in the IO group as compared to O ($p = 0.026$) and TO ($p = 0.006$), respectively. Furthermore, the isokinetic flexion peak torque (Fig. 4d) was found to be highest in the TO group compared to the O group (~1.1-fold, $p = 0.046$) and IO group (~1.5-fold, $p < 0.001$). Muscle volume, measured in the upper leg by MRI, was not significantly different between the three groups ($p > 0.05$, Fig. 4e), which was in accordance with the observed FFM.

**Higher exercise efficiency in trained older adults.** Resting energy expenditure was comparable across the three older adult groups ($p > 0.05$, Fig. 4f) and RER was also similar between the groups ($p > 0.05$, Fig. 4g). Furthermore, during the 1-h submaximal cycling test at 50% of W$_{max}$, no differences were observed in RER in TO as compared to O and IO ($p > 0.05$, Fig. 4g). TO showed a higher gross exercise efficiency in comparison to O ($p = 0.001$) and in comparison with IO ($p < 0001$, Fig. 4h) Likewise, the net exercise efficiency was higher in TO compared to both the O individuals ($p = 0.006$) and the IO group ($p = 0.002$, Fig. 4h).

**Higher whole-body insulin sensitivity in trained older adults.** Glucose infusion rate during the clamp ($M$-value) was ~1.3-fold higher in TO when compared to O ($p = 0.023$, Fig. 5a), indicating a higher whole-body insulin sensitivity in TO. Peripheral insulin sensitivity ($S_i$) was also higher in TO compared to O when expressed per kg body weight, which tended to be significant (TO: 0.066 (0.025) vs. O: 0.050 (0.028) μmol kgBW min$^{-1}$,

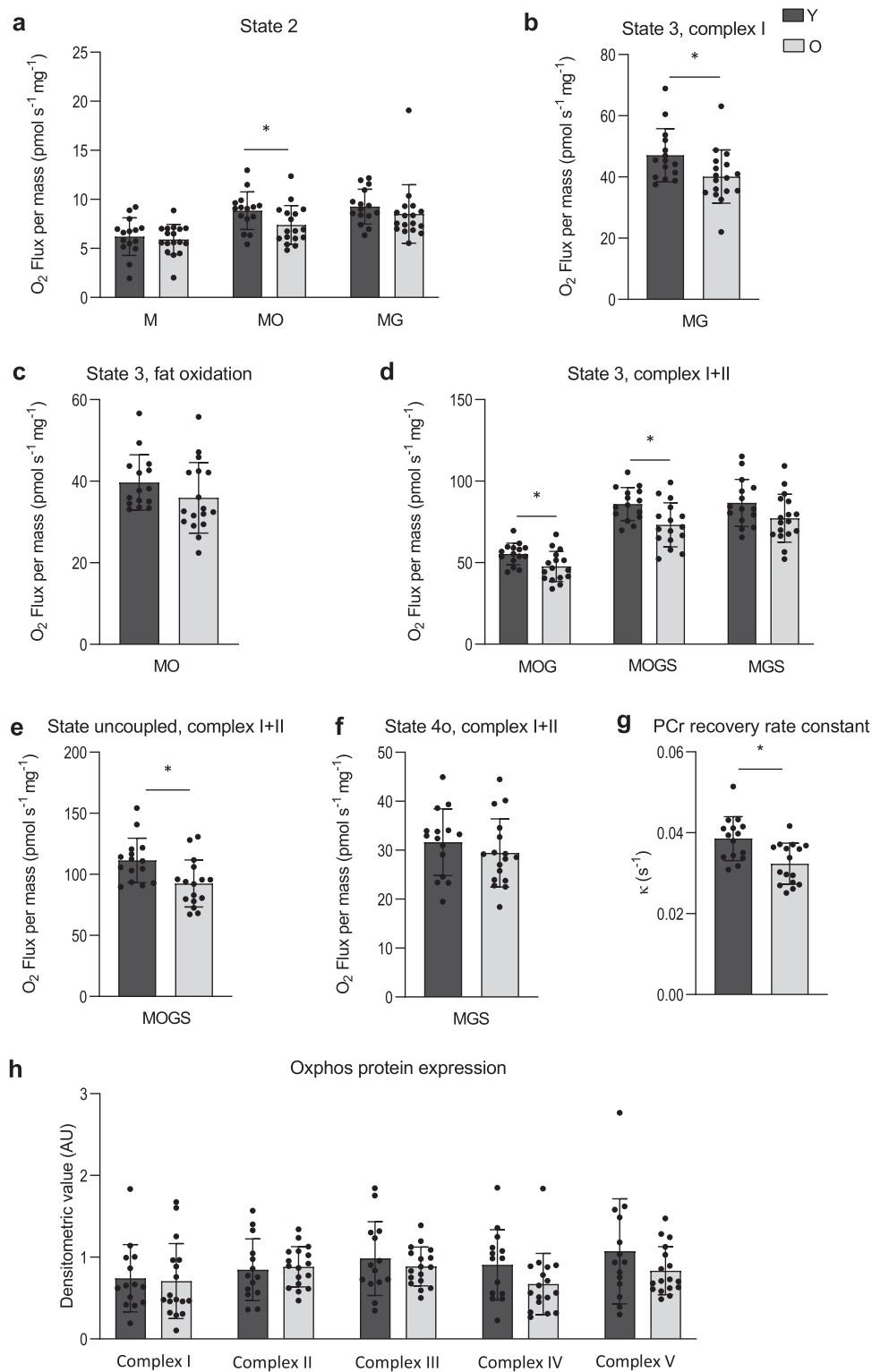

respectively, $p = 0.072$). When $S_i$ was corrected for FFM, no significant difference was observed between TO and O ($p = 0.165$, Fig. 5b). Insulin-mediated suppression of EGP (Fig. 5c) was similar in TO and O. Also RER was comparable between TO and O, both in the basal state ($p = 0.839$, Fig. 5d) and upon insulin stimulation ($p = 0.353$, Fig. 5d). As a result, metabolic flexibility was also comparable between TO and O (0.11 (0.04) vs. 0.10 (0.05), respectively, $p = 0.261$).

**Comparable gait stability in normally active and trained older adults.** When comparing O, TO, and IO, no significant differences were observed for gait variability during unperturbed walking (Supplementary Figs. 3 and 4). During the gait stability assessments, when considering the entire perturbation trial (Supplementary Fig. 5a), MoS was not significantly different between O and TO for perturbation 1, 2, and 9, indicating no significant stability or stepping behavior difference between O

**Fig. 3 Effect of aging on skeletal muscle mitochondrial function and density. a** Mitochondrial respiration upon substrates only (state 2) (Y, $n = 15$; O = 17). **b** ADP-stimulated respiration (state 3) respiration fueled by Complex I-linked substrates (Y, $n = 15$; O = 17). **c** State 3 respiration upon a lipid substrate (Y, $n = 15$; O = 17). **d** State 3 respiration upon parallel electron input into Complex II and I (Y, $n = 15$; O = 17). **e** Maximal uncoupled respiration upon FCCP (Y, $n = 15$; O = 17). **f** Mitochondrial leak respiration (state 4o) (Y, $n = 15$; O = 17). **g** $^{31}$P-MRS in vivo assessment of mitochondrial oxidative capacity (κ) (Y, $n = 15$; O, $n = 16$). **h** Mitochondrial protein expression of oxidative phosphorylation (OXPHOS) complex I, complex II, complex III, complex IV, and complex V (Y, $n = 15$; O, $n = 17$). Dark gray bars represent the young individuals (Y); light gray bars represent older individuals (O). In one Y no biopsy was performed due to technical issues and, in another Y, due to the implications of the SARS-CoV-19 outbreak. For the latter reason, also PCr recovery data is missing from the same Y participant. PCr data from one O has been excluded from analysis due to a pH decline below 6.9. PCr data from a Y participant has been excluded due to issues regarding the analysis. Values are presented as mean ± SD (with individual data points), Asterisk denotes significant differences between the two groups ($p < 0.05$, two-sided, independent samples $t$-test). M malate, O octanoyl-carnitine, G glutamate, S succinate, κ phosphocreatine resynthesis rate constant.

**Table 2 Participant body composition characteristics, physical activity, and physical function.**

|  | Older adults, normal physical activity (O) | Trained older adults (TO) | Physically impaired older adults (IO) |
|---|---|---|---|
| N | 17 | 19 | 6 |
| Gender F/M[a] | 8/9 | 8/11 | 3/3 |
| Age (years) | 71 (4)[b] | 68 (2)[c] | 71 (5)[b,c] |
| BMI (kg m$^{-2}$) | 25.8 (3.3)[b] | 23.6 (1.9)[c] | 27.0 (2.3)[b] |
| Body weight (kg) | 74.0 (11.9) | 68.3 (8.8) | 79.0 (7.2) |
| Fat mass (%) | 33.1 (9.2) | 26.2 (7.7)[c] | 37.4 (9.0)[b] |
| Fat mass (kg) | 24.5 (7.9)[b] | 17.6 (7.1)[c] | 29.4 (7.9)[b] |
| Fat-free mass (kg) | 49.5 (10.8) | 50.7 (9.7) | 49.6 (10.8) |
| Steps/day | 9983 (2781) | 13,815 (5934)[c] | 6608 (1765)[b] |
| HPA time (% of wake time) | 2.2 (1.2)[b] | 5.3 (3.9)[c] | 1.0 (0.6)[b] |
| LPA time (% of wake time) | 11.7 (2.6)[b] | 10.8 (2.9) | 8.3 (1.5)[c] |
| SPPB 4 m walk speed (m s$^{-1}$) | 1.1 (0.2) | 1.3 (0.2)[b] | 1.0 (0.3)[c] |
| SPPB Chair-stand test (s) | 10.1 (1.4) | 9.0 (2.1)[b] | 13.30 (4.8)[c] |

Values are presented as mean (SD). Age and HPA time were analyzed by two-sided Kruskal-Wallis tests followed by Bonferroni correction; other data were analyzed by one-way ANOVA with Tukey's post-hoc test.
*BMI* body mass index, *HPA* high-intensity physical activity, *LPA* low-intensity physical activity, *SPPB* short physical performance battery.
[a]Sex distribution across groups was tested by $\chi^2$ tests ($p = 0.982$).
[b,c]Groups that do not share the same letter are significantly different from each other ($p < 0.05$).

and TO. More detailed results regarding the gait variability, gait stability, and adaptability can be found in the supplementary notes and accompanying figures.

**Higher mitochondrial function in trained older adults**. In permeabilized muscle fibers, mitochondrial state 2 respiration was significantly higher in the TO individuals compared to O on all of the substrate combinations studied (Fig. 6a, $p < 0.05$). Also, ADP-stimulated (state 3) respiration on complex I-linked substrates as well as a lipid substrate was higher in TO compared to O. Thus, state 3 respiration fueled by malate + glutamate (MG) and malate + octanoylcarnitine (MO) were ~1.2-fold and ~1.4-fold higher in TO as compared to O ($p = 0.049$ and $p < 0.001$, respectively (Fig. 6b–c). ADP-stimulated respiration upon parallel electron input to both Complex I and II was also higher in TO compared to O. Thus, state 3 respiration upon malate + octanoyl-carnitine + glutamate (MOG, $p = 0.001$), malate + octanoyl-carnitine + glutamate + succinate (MOGS, $p = 0.002$), and malate + glutamate + succinate (MGS, $p = 0.018$) was ~1.2–1.3-fold higher in TO compared to O (Fig. 6d). Maximal FCCP-induced uncoupled respiration was ~1.4-fold higher in TO compared to O (state 3u, $p = 0.001$, Fig. 6e). State 4o (leak) respiration was higher in the TO group compared to both O (~1.3-fold, $p = 0.006$, Fig. 14f) and IO individuals (~1.3-fold higher, $p = 0.026$, Fig. 6f). Across the different respiratory states, oxygen consumption in permeabilized muscle fibers derived from IO appeared to be comparable to the respiration rates observed in O. However, due to the small sample size and large variation of the IO group, the differences between the IO and TO groups did not reach statistical significance in all respiratory states except for state 4o.

The negligible increase in oxygen consumption upon cytochrome C (2.61 (1.85) % vs. 2.27 (2.26) % vs. 0.33 (1.54) %, in TO vs. O vs. IO, respectively) underscores the viability and quality of the muscle fibers and was similar between the study groups ($p > 0.05$). Similar to the observed mitochondrial respiration rates in permeabilized muscle fibers, the TO group also displayed a ~1.2-fold and ~1.3-fold higher PCr recovery rate constant as compared to O and IO, respectively, although these differences did not reach statistical significance (TO vs. O $p = 0.169$; TO vs. OI, $p = 0.135$; O vs. IO $p = 0.815$; Fig. 6g). The exercise-induced PCr depletion and pH was similar across the three study groups ($p > 0.05$) and pH remained above 6.9 indicating no substantial exercise-induced acidification. Protein expression of OXPHOS complex I and III was higher in TO compared to O ($p = 0.010$ and $p = 0.042$, respectively, Fig. 6h). Furthermore, protein expression of complex II ($p = 0.003$ and $p = 0.025$, respectively) and complex IV ($p < 0.001$ and $p = 0.003$, respectively) was higher in TO compared to both O and IO.

**Mitochondrial and muscle function and exercise efficiency are related**. To further explore the relationship between mitochondrial energetics, muscle quality and physical function, correlations were performed based on the combined data from the entire study cohort. Pearson correlation outcomes are summarized in Supplementary Table 2 and illustrated in Fig. 7. As anticipated, correlations were observed between the different measures for mitochondrial function and oxidative capacity (ex vivo respiration, in vivo PCr recovery rate and VO$_2$max, Fig. 7a). Furthermore, measures for ex vivo mitochondrial capacity – but not in vivo PCr recovery rate constant – were found to significantly

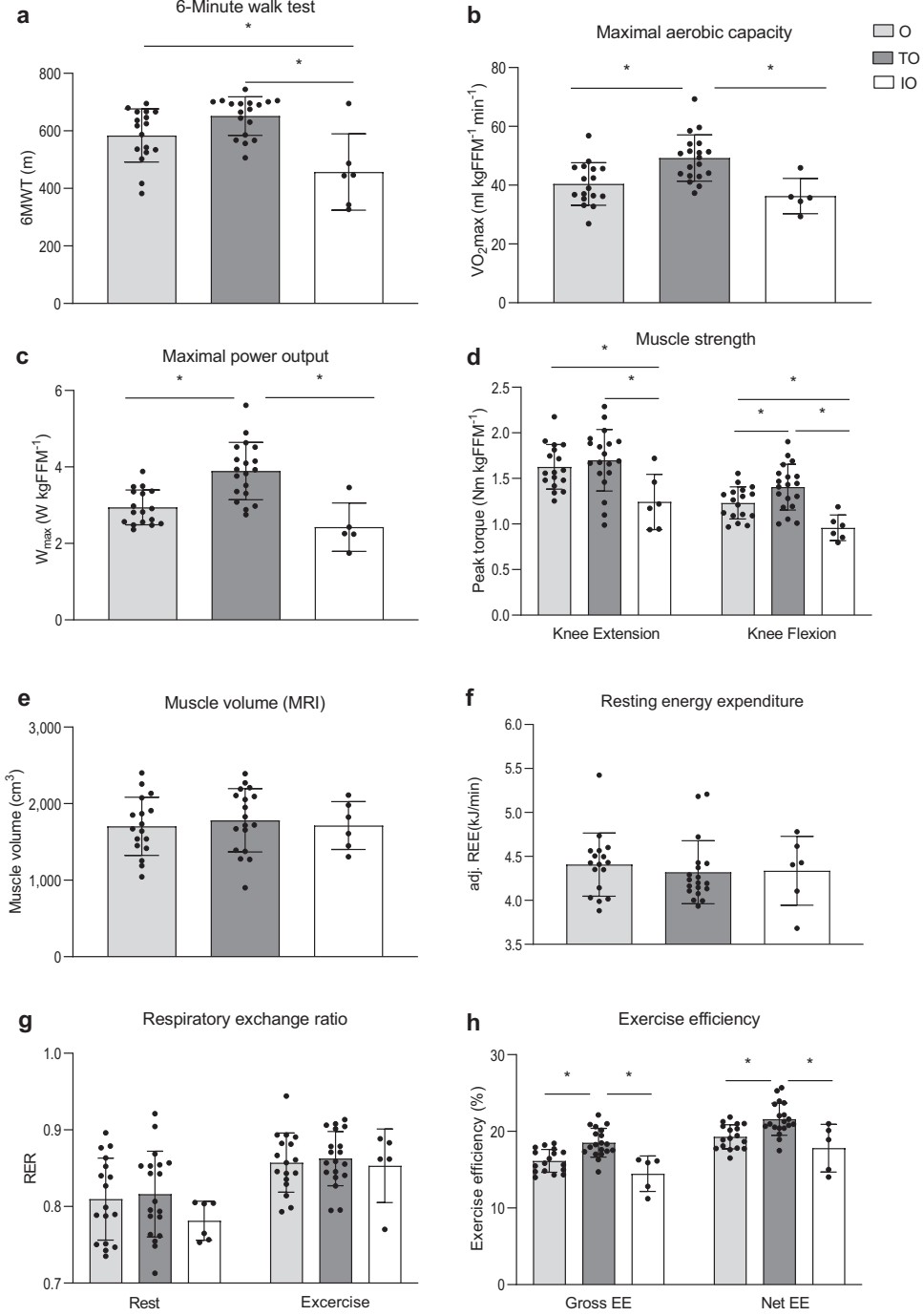

**Fig. 4 Effect of exercise training and physical impairment on muscle health and exercise efficiency in older adults. a** Walking distance during the 6MWT performed on the Caren-system (O, $n = 17$; TO, $n = 17$; IO, $n = 6$). **b** Maximum rate of oxygen consumption measured during graded cycling exercise (O, $n = 17$; TO, $n = 19$; IO, $n = 6$). **c** Maximal power output measured during graded cycling test (O, $n = 17$; TO, $n = 19$; IO, $n = 6$). **d** Muscle strength expressed as the extension and flexion peak torque during an isokinetic protocol on the Biodex system and corrected for fat-free mass (O, $n = 17$; TO, $n = 19$; IO, $n = 6$). **e** Upper leg muscle volume measured by MRI (O, $n = 17$; TO, $n = 18$; IO, $n = 6$). **f** Resting energy expenditure adjusted for FFM (O, $n = 17$; TO, $n = 19$; IO, $n = 5$). **g** Respiratory exchange ratio measured before, in resting conditions, and during submaximal exercise (O, $n = 17$; TO, $n = 19$; IO, $n = 5$). **h** Exercise efficiency measured during a submaximal cycle test and expressed as gross efficiency and net efficiency (O, $n = 17$; TO, $n = 19$; IO, $n = 5$). Light gray bars represent normally active older adults (O); dark gray bars represent trained older adults (TO); white bars represent physically impaired older adults (IO). Upon the advice of the responsible medical doctor, one IO participant did not perform the maximal and submaximal exercise tests. One TO participant did not perform the 6MWT due to scheduling issues. The reported 6MWT distance from another TO was invalid and therefore excluded for analysis. Data presented in panels **a** and **c** were analyzed by two-sided Kruskal–Wallis tests followed by Bonferroni correction. Data presented in panels **b**, **d**–**h** were analyzed by one-way ANOVA with Tukey's post-hoc test. Values are presented as mean ± SD (with individual data points), asterisk denotes significant differences between two groups ($p < 0.05$). 6MWT 6-min walk test distance, $VO_2max$ maximal aerobic capacity, Nm newton meters, FFM fat-free mass, RER respiratory exchange ratio, REE resting energy expenditure, EE exercise efficiency.

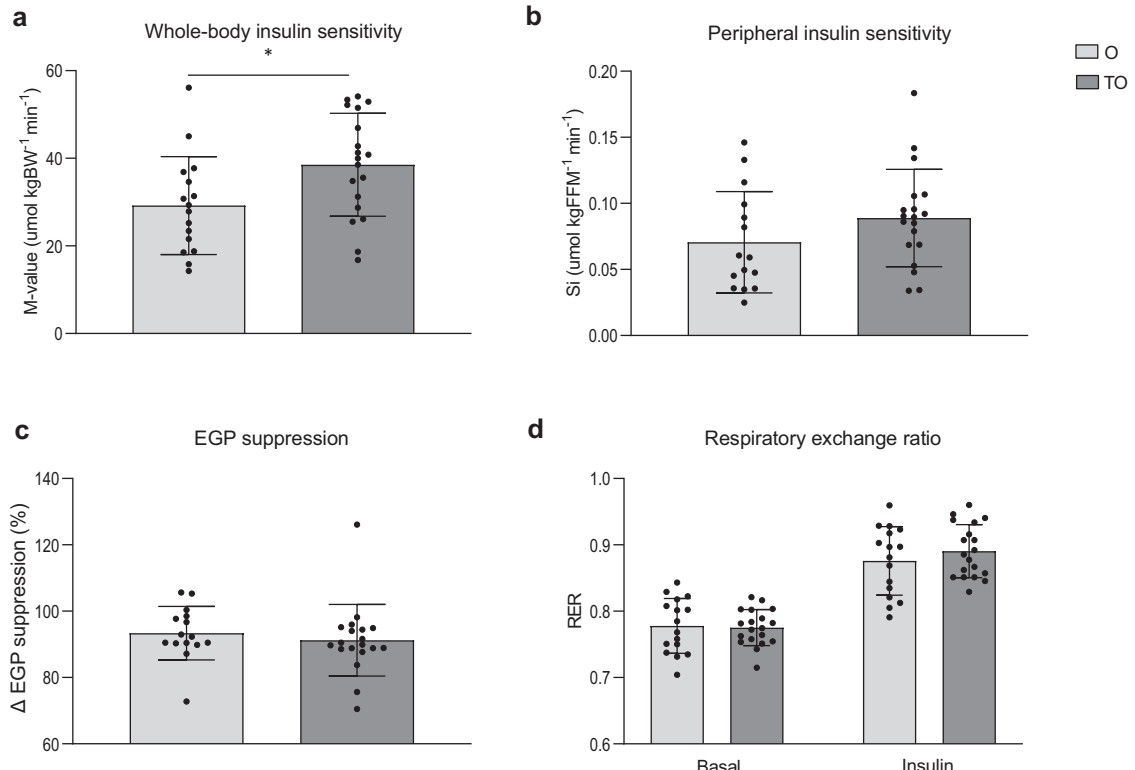

**Fig. 5 Effect of exercise training on insulin sensitivity and substrate selection in older adults. a** Whole-body insulin sensitivity based on the *M*-value measured during a hyperinsulinemic-euglycemic clamp. **b** Insulin-stimulated glucose uptake corrected for plasma insulin and glucose levels ($S_i$) and expressed per kg FFM **c** EGP suppression, calculated as the percentage insulin-suppressed EGP from the basal EGP. **d** Respiratory exchange ratio measured before, in resting conditions (Basal), and during insulin stimulation (Insulin). Light gray bars represent normally active older adults (O, $n = 15$); dark gray bars represent trained older adults (TO, $n = 19$). Because only 3 IO participants could undergo the clamp, these results were not considered. One O participant was excluded for analysis due to violation of the protocol instructions. For one O participant tracer data could not be analyzed. Values are presented as mean ± SD (with individual data points), asterisk denotes significant differences between the two groups ($p < 0.05$ two-sided, independent samples *t*-test). *M*-value mean glucose infusion rate, BW body weight, $S_i$ insulin-stimulated glucose disposal, FFM fat-free mass, EGP endogenous glucose production.

correlate with 6MWT, *M*-value and steps per day as markers for walking performance, insulin sensitivity, and physical activity, respectively (Supplementary Table 2 and Fig. 7b, c). Both step length variability and double support time variability correlated with measures of in vivo and ex vivo mitochondrial function (Supplementary Table 2 and Fig. 7e, f). Additionally, ex vivo mitochondrial coupled respiration rates were found to correlate significantly with gross exercise efficiency (Supplementary Table 2 and Fig. 7d). Interestingly, gross and net exercise efficiency showed the strongest correlations with multiple aspects of muscle health including endurance, physical function, muscle strength, insulin sensitivity, gait stability, and physical activity, as illustrated by the significant correlations with VO₂max, 6MWT, the chair-stand test, isokinetic extension, *M*-value, $S_i$, recovery steps after perturbation, and steps per day (Supplementary Table 2 and Fig. 7g, h), respectively.

Partial correlation analyses adjusted for age, sex, and BMI are shown in Table 3. Also, after adjustment, correlations were observed between the different measures for mitochondrial function and oxidative capacity (ex vivo respiration, in vivo PCr recovery rate, and VO₂max).

Interestingly, maximal ex vivo mitochondrial respiration rates were found to significantly correlate with steps per day, a marker for PA. In addition, ex vivo mitochondrial capacity – but not in vivo PCr recovery rate constant – were significantly correlated with gait stability assessed by double step support variability and insulin sensitivity as assessed by the *M*-value. Moreover, ex vivo

mitochondrial uncoupled respiration rates were found to correlate significantly with gross exercise efficiency. Interestingly, gross and net exercise efficiency showed, also after adjustment, the strongest correlations with multiple aspects of muscle health including endurance, physical function, insulin sensitivity, and PA. The correlation between exercise efficiency and step variability was no longer present after adjustment for age and BMI.

## Discussion

Skeletal muscle mitochondrial function has been shown to decline with age and may underlie the decline in muscle health and performance during aging. Although the origins of reduced mitochondrial function with age are a matter of ongoing debate, recent data suggest that PA levels may be a key determinant of mitochondrial energetics in aging[6]. Although the data presented here support a role for PA in mitigating the negative effects of sedentary behavior in older adults, we nevertheless show that older adults with normal PA when compared to young adults with equivalent habitual PA levels, display lower mitochondrial capacity despite comparable mitochondrial content. Additionally, the older participants had lower muscle strength, aerobic capacity, exercise efficiency, and gait stability compared to similarly active young adults. Conversely, trained older participants exhibited higher mitochondrial capacity than older adults with normal PA and older adults with impaired physical function, which was associated with improved functional outcomes.

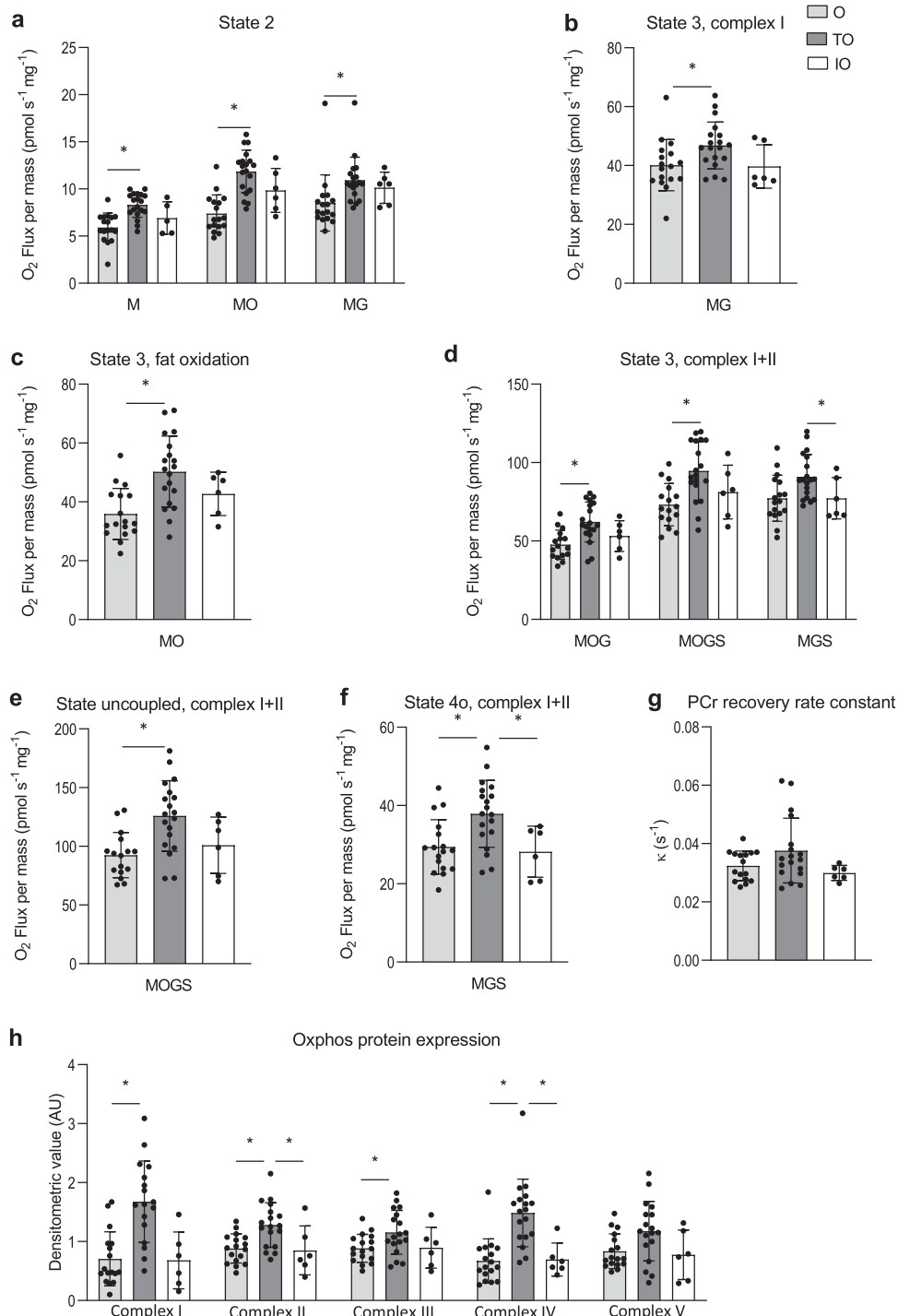

**Fig. 6 Effect of exercise training and physical impairment on skeletal muscle mitochondrial respiration and density in older adults. a** Mitochondrial respiration upon substrates only (state 2) (O, $n = 17$; TO, $n = 19$; IO, $n = 6$). **b** ADP-stimulated respiration (state 3) fueled by Complex I-linked substrates (O, $n = 17$; TO, $n = 19$; IO, $n = 6$). **c** State 3 respiration upon a lipid substrate (O, $n = 17$; TO, $n = 19$; IO, $n = 6$). **d** State 3 respiration upon parallel electron input into Complex I and II (O, $n = 17$; TO, $n = 19$; IO, $n = 6$). **e** Maximal uncoupled respiration upon FCCP (O, $n = 17$; TO, $n = 19$; IO, $n = 6$). **f** Mitochondrial leak respiration (state 4o) (O, $n = 17$; TO, $n = 19$; IO, $n = 6$). **g** In vivo $^{31}$P-MRS estimate of mitochondrial oxidative capacity $\kappa$ (O, $n = 16$; TO, $n = 18$; IO, $n = 6$). **h** Mitochondrial protein expression of oxidative phosphorylation (OXPHOS) complex I, complex II, complex III, complex IV, and complex V (O, $n = 17$; TO, $n = 18$; IO, $n = 6$). Light gray bars represent normally active older adults (O); dark gray bars represent trained older adults (TO); white bars represent physically impaired older adults (IO). PCr data from one O participant has been excluded from analysis due to a pH decline below 6.9. PCr from a TO participant has been excluded due to issues regarding the analysis. From one TO not enough muscle tissue was available for the OXPHOS analysis. Data presented in panels **a** and **g** were analyzed by two-sided Kruskal–Wallis tests followed by Bonferroni correction. Data presented in panels **b**–**f** and **h** were analyzed by one-way ANOVA with Tukey's post-hoc test. Values are presented as mean ± SD (with individual data points), asterisk denotes significant differences between two groups ($p < 0.05$). M malate, O octanoyl-carnitine, G glutamate, S succinate, $\kappa$ phosphocreatine resynthesis rate constant.

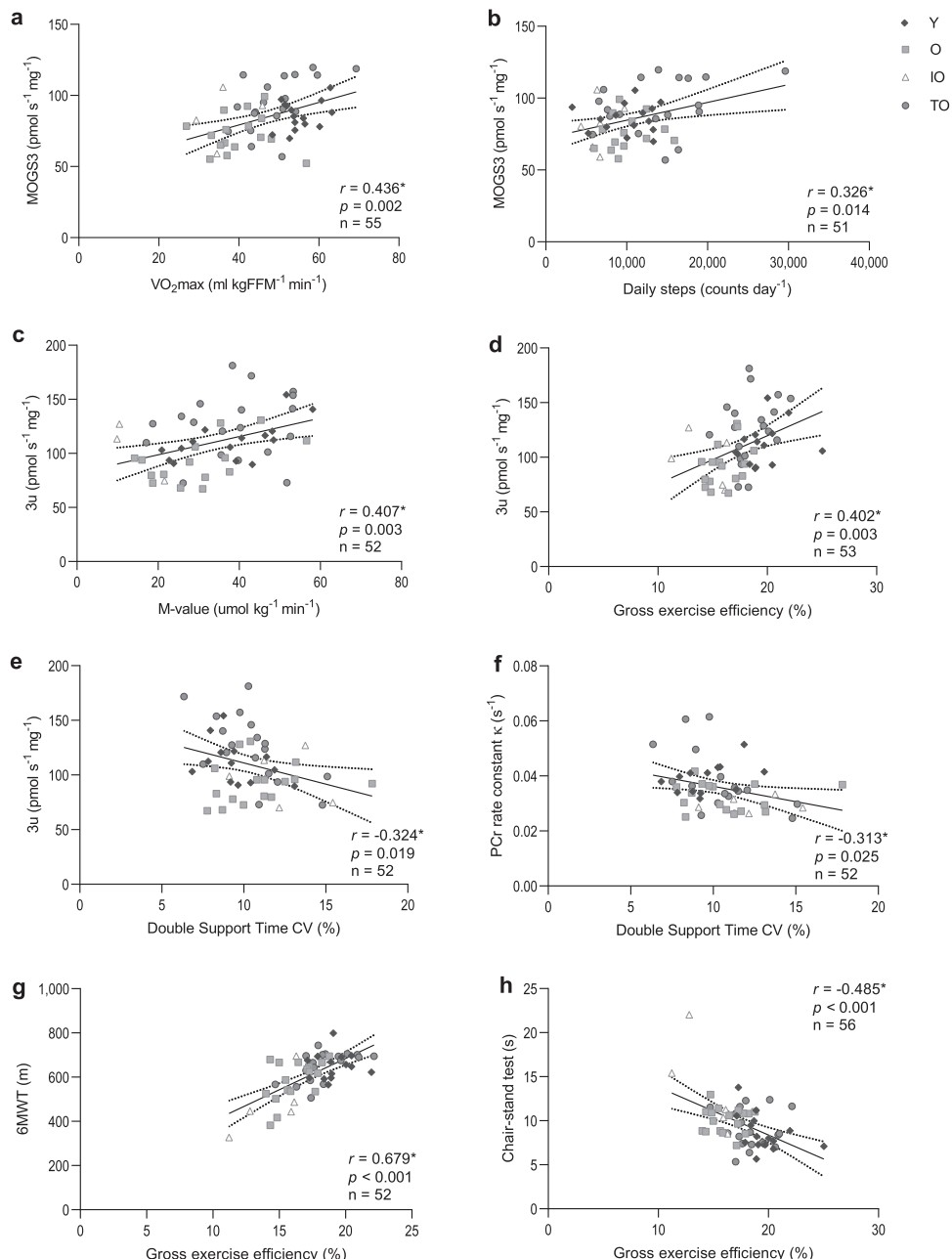

**Fig. 7 Bivariate correlations between mitochondrial function and physical muscle function. a** Correlation between maximal exercise capacity and maximal coupled mitochondrial respiration. **b** Correlation between physical activity and maximal coupled mitochondrial respiration. **c** Correlation between whole-body insulin sensitivity and maximal uncoupled mitochondrial respiration. **d** Correlation between gross exercise efficiency and maximal uncoupled mitochondrial respiration. **e** Correlation between step variability and maximal uncoupled mitochondrial respiration **f** Correlation between step variability and in vivo PCr recovery rate constant. **g** Correlation between walking performance and gross exercise efficiency. **h** Correlation between SPPB chair-stand test and gross exercise efficiency. Dark gray diamonds indicate young normally active individuals (Y), light gray squares indicate normally active older adults (O), white triangles indicate physically impaired older adults individuals (IO), and dark gray circles indicate trained older adults (TO). Asterisk indicates the correlation is significant at the 0.05 level (2-tailed, $p < 0.05$); Best-fit trend line and 95% confidence intervals are included. MOGS3 state 3 respiration upon malate + octanoyl-carnitine + glutamate + succinate, 3u state 3 uncoupled respiration upon FCCP, κ phosphocreatine resynthesis rate constant, 6MWT 6-min walk test, VO2max maximal aerobic capacity, FFM fat-free mass, Nm newton meters, *M*-value mean glucose infusion rate, r Pearson correlation coefficient.

Notably, the increase in mitochondrial capacity in the trained older adults was largely accounted for by an increase in mitochondrial content. These data suggest a role for aging-associated reductions in mitochondrial function independent of PA levels or mitochondrial content and may be a contributory factor in physical function decline. These effects appear to be mitigated by exercise training-induced increases in mitochondrial content.

In this study, the young and older adult groups were not engaged in structured exercise activities and both recorded ~10,000 steps daily. This is well above the average daily step count reported for older untrained adults in previous studies[6,26] and indicates an overall active lifestyle and adequate levels of PA for a healthy adult[27]. Despite similar habitual PA, walking perfor-mance and maximal aerobic capacity was lower in the older

**Table 3 Partial analysis to explore the relationships between energetics and physical muscle function.**

| Measurement | Variables | | 6MWT (m) | Chair-stand test (s)[a] | VO₂max (ml min⁻¹ kgFFM⁻¹) | Isokinetic extension (Nm kgFFM⁻¹) | Isokinetic flexion (Nm kgFFM⁻¹) | M-value (µmol kg⁻¹ min⁻¹) | $S_i$ (µmol kgFFM⁻¹ min⁻¹) | Daily steps (counts day⁻¹) | Step length variability[c] | Double support time variability[c] | Gross efficiency (%) |
|---|---|---|---|---|---|---|---|---|---|---|---|---|---|
| Mitochondrial function | Maximal coupled respiration MOGS3 (pmol s⁻¹ mg⁻¹) | r | 0.165 | −0.084 | 0.416[b] | 0.049 | −0.054 | 0.253 | 0.062 | 0.363[b] | −0.094 | −0.293[b] | 0.232 |
| | | p | 0.256 | 0.548 | 0.002 | 0.727 | 0.699 | 0.080 | 0.673 | 0.009 | 0.534 | 0.048 | 0.101 |
| | Maximal uncoupled respiration (pmol s⁻¹ mg⁻¹) | r | 0.109 | −0.140 | 0.512[b] | 0.077 | −0.036 | 0.317[b] | 0.169 | 0.422[b] | −0.170 | −0.378[b] | 0.304[b] |
| | | p | 0.458 | 0.318 | <0.001 | 0.584 | 0.798 | 0.026 | 0.250 | 0.002 | 0.258 | 0.010 | 0.030 |
| | PCr recovery rate constant κ (s⁻¹) | r | 0.191 | −0.037 | 0.363[b] | 0.035 | −0.024 | 0.238 | 0.124 | 0.305[b] | −0.216 | −0.271 | 0.233 |
| | | p | 0.194 | 0.795 | 0.009 | 0.807 | 0.866 | 0.103 | 0.406 | 0.032 | 0.150 | 0.068 | 0.103 |
| Exercise efficiency | Gross efficiency (%) | r | 0.597[b] | −0.478[b] | 0.570[b] | 0.110 | 0.052 | 0.424[b] | 0.224 | 0.361 | −0.132 | −0.152 | – |
| | | p | <0.001 | <0.001 | <0.001 | 0.434 | 0.710 | 0.002 | 0.121 | 0.009 | 0.383 | 0.315 | – |
| | Net efficiency (%) | r | 0.521[b] | 0.418[b] | 0.470[b] | 0.003 | −0.040 | 0.397[b] | 0.235 | 0.273 | −0.103 | −0.215 | – |
| | | p | <0.001 | 0.002 | <0.001 | 0.983 | 0.778 | 0.004 | 0.104 | 0.052 | 0.494 | 0.152 | – |

Data are from all groups (total $n > 51$).
Correlations are adjusted for [a] age, sex, and BMI.
[a] indicates variables that were log transformed.
[b] indicates the partial correlation is significant at the 0.05 level (two-sided, $p < 0.05$).
[c] indicates the correlations corrected for only [a] age and BMI.
MOGS3 state 3 respiration upon malate + octanoyl-l-carnitine + glutamate + succinate, 6MWT 6-min walk test, VO₂max maximal oxygen flow, FFM fat-free mass, Nm newton meters, M-value mean glucose infusion rate, $S_i$ insulin-stimulated glucose uptake, r partial correlation coefficient.

adults compared to the young controls. These findings are in line with previous observation from a study comparing adults <65 years and >65 years, which observed the peak VO₂ and 400 m gait speed to be lower with higher age[5].

With an average of more than 12,500 steps per day and at least three exercise sessions per week, the trained older participants in the current study are considered to be highly physically active, whereas physically impaired older adults averaged <7500 steps per day, reflecting a sedentary lifestyle[27]. The differences in habitual PA levels between the groups of older adults were reflected in walking performance and maximal aerobic capacity, confirm previous evidence[6,20] and support the notion that at advanced age, increasing PA by exercise training can help to preserve physical function and performance.

We could not detect a statistical difference in muscle volume between young and older adults with similar levels of PA, indicating that habitual PA may be an important factor in the age-associated loss in muscle volume that has been observed previously[28]. In our study, however, training status did not affect muscle volume in older adults, although this may be explained by the fact that the cohort consisted primarily of endurance-trained athletes and endurance training is less likely to improve lean mass compared to strength training[29]. Despite the lack of difference in muscle volume, muscle strength was lower in older participants and differences in muscle strength corresponded to habitual PA stratification applied to the three older adults groups, which is in agreement with previous comparisons between active and sedentary older adults[6,30].

Exercise efficiency refers to the energetic cost of physical activities such as walking and decreases in exercise efficiency likely contribute to the age-related inefficiency of locomotion and daily activities[21]. As expected, lower net exercise efficiency was observed in the older participants compared to young adults in the present study, underlining an increased energy expenditure during exercise in the presence of comparable resting metabolic rate. This observation is in line with the findings from Distefano et al.[6], showing a lower exercise efficiency in older adults when compared to young adults with comparable levels of PA (~8500 vs. ~10,500 steps per day, respectively).

The relative difference of ~15% in exercise efficiency in the trained older adults compared to the older adults with normal PA are similar to two previous studies suggesting that exercise training interventions have the potential to reverse age-related reductions by increasing exercise efficiency up to ~30% in sedentary[31] and endurance-trained volunteers[32]. These findings are further supported by a cross-sectional study observing a higher exercise efficiency in athletic adults when compared to sedentary older adults[20]. Interestingly, mitochondrial dysfunction has been implicated as a key factor in the reduced exercise efficiency with age which indicates mitochondria as the target for intervention to reverse this source of debilitation in older adults[20,21]. We observed a strong positive correlation between maximal mitochondrial capacity and gross exercise efficiency suggesting that mitochondrial capacity is indeed a determinant of exercise efficiency in aging.

Older adults responded in a less effective manner to the first perturbation when compared to the young participants, underlining the higher risk of falls in this population[7], even when PA is comparable. This observation confirms numerous other studies demonstrating age related differences in the ability to cope with such perturbations (for example:[33,34]). As PA was comparable between young and older adults in this study, it might be inferred that an increased fall risk with age cannot be entirely attributed to a decrease in PA, as is often suggested. During unperturbed walking, most spatiotemporal parameters and their variability were not affected by age, indicating that perhaps steady-state

walking is more readily preserved with PA. Nevertheless, this only seems to be confirmed to a certain extent, as the trained older adults in the current study did not have significantly different unperturbed gait outcomes, compared to the normally active older adults.

One previous study reported that in a mixed group of older and younger adults, those who were recreational runners performed better on a forward falling task than the non-active participants[35]. This pattern was not observed in the current study, as there were no clear differences between the TO and O groups on the perturbation tasks. It could be argued that this may be partly due to the fact that the O group actually had quite a high habitual PA but this did not seem to result in a lack of difference to the young adults. Another possibility is that the current gait perturbation task, which is more dynamic and in which it is practically impossible to predict and prepare for the perturbations, is simply more complex than the 'forward lean and release task' in such a way that high levels of PA do not benefit the performance of the task as much.

The ability to improve stability recovery following repeated gait perturbations was preserved in older adults with normal PA when compared to young adults. Furthermore, both the O and TO groups significantly reduced their required recovery steps by the ninth perturbation. These results confirm many previous reports of adaptability to gait perturbations being preserved in older age and reinforce the potential value of perturbation-based balance training in this population[36–39]. However, it does appear that exercise training in older participants does not lead to a better ability to adapt to perturbations. A recent review paper (82) highlighted that reduced adaptability to perturbations is mainly seen in populations with disrupted sensory and nervous systems. As the O and TO participants of the present study were in good health, adaptability may have already been sufficient in both groups, with little room for improvement through physical activity.

In the current study, we found that both in vivo PCr recovery rate constant and ex vivo mitochondrial respiration in skeletal muscle was lower in older compared to young adults. It is important to note that in the present study the decrease in mitochondrial function has been observed in older adults with PA levels being well above the general recommendations[40]. These results suggest that maintaining a healthy level of PA is not sufficient to prevent an age-related decline in mitochondrial function. These findings are in line with the age-related decline in the mitochondrial ATP production rate (MAPR) observed in young and older adults who exercised <30 min a week[14,41]. An even more prominent age-related decline in MAPR was observed comparing young and older adults who were similarly endurance-trained[41], further hinting at an independent, aging-induced decline in skeletal muscle mitochondrial function. Nevertheless, findings of the present study are in contrast with previous observations by Distefano et al.[6,42], who found no chronological age effect on mitochondrial function in deeply phenotyped young and older adults. This discrepancy could be explained by gender differences, as we included nine male and eight female volunteers in both groups while Distefano et al. included eight male and two female participants in each group. Further research is needed to clearly define the effect of aging on skeletal mitochondrial function at different levels of PA and examine if that is different in male versus female volunteers[43].

Interestingly, ex vivo ADP-stimulated mitochondrial respiration in permeabilized muscle fibers was ~15% lower in older adults when compared to young adults, whereas trained older adults displayed a ~30% higher respiration in comparison to older adults with normal PA. These findings support the retainment of skeletal muscle plasticity in response to exercise with

aging and indicate that mitochondrial respiratory function can be preserved through exercise[6,41]. In line with the present study, other reports indicate that exercise intensity is an important determinant of ameliorating age-related biological functions since rigorous exercise programs (such as high-intensity interval training) – but not modest aerobic activity – was able to prevent age-related decline in mitochondrial respiration[44]. In contrast with ex vivo assessed mitochondrial respiration results, in vivo mitochondrial function, assessed by the $^{31}$PCr recovery rate constant, was not found to be different between the older adult groups. This discrepancy could be explained by the fact that $^{31}$P-MRS may be confounded by limiting systemic factors such as adequate perfusion and oxygen delivery during and after exercise[45], and other characteristics of the microenvironment. Interestingly, mitochondrial density, estimated by OXPHOS protein levels, appeared to be similar between young and older participants, indicating that the age-related loss in mitochondrial function is not explained by a lower mitochondrial mass. In trained older adults, however, OXPHOS proteins were higher, suggesting that increased levels of mitochondrial content do underlie the positive effect of exercise training on mitochondrial function in older individuals. These results suggest that aging and exercise training affect mitochondrial function via distinct mechanisms. We did not observe a difference in either ex vivo or in vivo skeletal muscle mitochondrial capacity in physically impaired older individuals as compared to healthy older adults with normal PA. Due to the small sample size of this group, however, these outcomes should be interpreted with caution.

To further explore the relation between mitochondrial oxidative capacity and multiple aspects of muscle health, we subjected the entire cohort ($n = 59$) to correlative analyses. We found that besides ex vivo mitochondrial capacity also in vivo mitochondrial function associated with maximal aerobic capacity and exercise efficiency, which further supports that mitochondrial energetics are likely important factors in loss of muscle function with aging. Similar results have recently been found by Distefano et al[6]. and Gonzalez-Freire et al[46]. Moreover, mitochondrial function correlated with measures of gait variability, indicating that mitochondrial function potentially influences the efficiency and consistency of the gait pattern. Interestingly, we found that exercise efficiency correlated strongly with mitochondrial respiratory capacity, as well as with physical function parameters such as the 6MWT and the chair-stand test. Taken together, our data support a role for mitochondrial capacity and in maintaining muscle function and indicate that exercise efficiency may be a good proxy for skeletal muscle mitochondrial capacity and muscle health.

The isolated impact of aging on skeletal muscle is difficult to unravel from the many other factors that change concurrently with aging, including decreased PA and increased adiposity[47]. Despite similar levels of habitual of PA in the current study, body fat was higher and (whole-body) insulin sensitivity was lower in the older participants when compared to young. In comparison with older adults with normal PA, trained older adults displayed higher levels of whole-body insulin sensitivity, underscoring the ability of exercise training to improve insulin sensitivity and glucose uptake, also at an older age. However, since young and trained older adults also featured a lower adiposity, differences in body composition may (partly) mediate the observed differences in insulin sensitivity, independent of mitochondrial function. Increased adiposity in older participants has also been negatively associated with muscle function, independent of muscle mass[48]. Furthermore, unhealthy aging is characterized by an augmented accumulation of intramyocellular lipids[6,26], which could influence muscle function[6] and insulin sensitivity[49]. Moreover, differences in BMI and body composition, inherent to the different study

group characteristics, have been suggested to also influence skeletal muscle mitochondrial capacity[42,50]. Taken together, our data indicate that body fat accumulation with increasing age may be a contributing factor in potentiating the loss of muscle quality in older adults and suggest that high-intensity physical activity may be necessary to minimize changes in body composition over the lifespan and to mitigate muscle aging.

Our results suggest that, despite maintaining an adequate physical activity level, aging is associated with low in vivo and ex vivo mitochondrial capacity, maximal aerobic capacity, exercise efficiency, gait stability, muscle strength, insulin sensitivity, physical function, and increased body fat. Nevertheless, increasing physical activity through regular exercise training partially protects against these age-related declines in (mitochondrial) oxidative capacity and muscle health. Finally, we showed that mitochondrial capacity was positively associated with exercise efficiency and insulin sensitivity, supporting the idea that mitochondria represent a promising therapeutic target to negate the aging-associated deterioration of skeletal muscle health in order to preserve physical function and performance.

## Methods

**Participants**. Fifty-nine participants, including 17 young (9 male and 8 female) and 42 older individuals (23 male and 19 female), were recruited in the community of Maastricht and its surroundings through advertisements placed on the Maastricht University campus, in newspapers, supermarkets, and at sports clubs. The study was conducted in accordance with the principles of the declaration of Helsinki and approved by the Ethics Committee of the Maastricht University Medical Center+. All participants provided their written informed consent, and the study was registered at clinicaltrials.gov with identifier NCT03666013.

Prior to inclusion, all participants underwent a medical screening that included a medical questionnaire, a physical examination by a physician, and an assessment of physical function by means of the Short Physical Performance Battery, which includes a standing balance test, a 4-m walk test, and a chair-stand test. The SPPB score was calculated according to the cut-off points determined by Guralnik et al.[51]. A sitting blood pressure measurement and an electrocardiogram (ECG) were also performed. After the screening procedure, participants were assigned to the following study groups: young individuals with normal physical activity (Y, 20–30 years), older adults with normal physical activity (O, 65–80 years), trained older adults (TO, 65–80 years), and physically impaired older adults (IO, 65–80 years). Participants were considered normally physically active if they completed no more than one structured exercise session per week, whereas participants were considered trained if they engaged in at least three structured exercise sessions of at least 1 h each per week for an uninterrupted period of at least the past year. Participants were classified as older adults with impaired physical function (IO) in case of an SPPB score of ≤9. Upon inclusion, further details on habitual physical activity levels were obtained during the study using accelerometry (ActivPAL).

Exclusion criteria were contra-indications for MRI examination, uncontrolled hypertension, the use of medication that could interfere with the results of the study, medical history of cardiovascular disease, type 2 diabetes mellitus, or other health problems that may hamper the safety of the individual during participation. Impairments in parameters of liver and kidney function were examined via determination of plasma aspartate aminotransferase (ASAT), alanine aminotransferase (ALAT), gamma-glutamyltransferase (γGT), bilirubin, and creatine.

**Experimental design**. This cross-sectional study was conducted at the Maastricht University Medical Center+, The Netherlands, between September 2017 and March 2020.

The detailed assessment of muscle health consisted of various measurements divided over five study visits, equally distributed over a period of 5 weeks to allow for sufficient recovery between each visit and prevent interference between the various measurements (Supplementary Table 1). During their participation, participants were instructed to maintain their habitual diet and physical activity pattern, and in the 3 days preceding the test days, participants refrained from strenuous physical activity.

**Hyperinsulinemic-euglycemic clamp**. A hyperinsulinemic-euglycemic clamp was performed to assess peripheral insulin sensitivity, as previously described[52]. Briefly, participants reported to the laboratory at 8:00 a.m. after an overnight fast from 10:00 p.m. A fasted blood sample was taken and subsequently, a primed-continuous infusion of d-[6,6-2H2]-glucose ($0.04\,\mathrm{ml\,kg^{-1}\,min^{-1}}$) was initiated[53]. After 3 h ($t = 180\,\mathrm{min}$), infusion of insulin ($40\,\mathrm{mU\,m^{-2}\,min^{-1}}$) was started for a period of 2.5 hours. Based on the continuously monitored plasma glucose concentrations, the glucose infusion rate (GIR) was adjusted to maintain a steady-state

plasma glucose level of ~5.0 mmol L$^{-1}$. Subsequently, whole-body insulin sensitivity was determined by calculating the $M$-value ($\mathrm{mg\,kg^{-1}\,min^{-1}}$). During the last 30 min of basal sampling ($t = 150{-}180\,\mathrm{min}$) and during the last 30 min of the insulin infusion ($t = 300{-}330\,\mathrm{min}$), blood samples were collected and indirect calorimetry (Omnical, IDEE, Maastricht, The Netherlands) was performed. Based on the measured oxygen and carbon dioxide concentrations, substrate oxidation rates were calculated using equations with the assumption that protein oxidation was negligible[54]. Steele's single pool non-steady-state equations were used, allowing small differences in glucose concentrations to calculate glucose appearance (Ra) and (insulin-stimulated) glucose disposal (Rd)[53]. The change in Rd (from baseline to clamp steady-state) was corrected for plasma insulin and glucose levels during the clamp (S$_i$), as described in Remie et al.[52]. Endogenous glucose production (EGP) was calculated as Ra minus exogenous glucose infusion rate.

Three individuals in the IO group were excluded from participation in the clamp on the grounds of safety. As a result, the small number ($n = 3$) remaining made a comparison unviable and we therefore excluded the IO group from the final insulin sensitivity comparisons.

**Maximal aerobic capacity**. Maximal aerobic capacity (V̇O₂max) was assessed with concurrent ECG during a graded cycling test until exhaustion, as described previously[55]. Briefly, after a warming-up period of 5 min at 50 Watt, the power output was increased every 2.5 min by 50 Watt until levels above 80% of the predicted maximal heart rate (=220 years – age) were observed. After this point, the test continued, and power output was increased every 2.5 min by 25 Watt until the participant could no longer pedal above 60 revolutions per minute. Consumed O₂ and expired CO₂ were measured continuously throughout the test using indirect calorimetry (Omnical, IDEE, Maastricht, The Netherlands) to determine V̇O₂max.

**Habitual physical activity**. Habitual physical activity was determined using an ActivPAL3 monitor (PAL Technologies, Glasgow, Scotland) for a consecutive period of 5 days, including two weekend days. The monitor was wrapped and attached to the skin on the anterior aspect of the upper leg using Tegaderm (3M™) in a waterproof fashion; non-wear was therefore not an issue. Data were uploaded using the software provided by ActivPAL and processed using customized software written in MATLAB R2013b (MathWorks, Natick, MA, USA). Besides the total amount of steps per day, the total stepping time was calculated in proportion to waking time per day. Stepping time (i.e. physical activity) was further classified into high-intensity physical activity (HPA; minutes with a step frequency >110 steps min$^{-1}$ in proportion to waking time) and lower-intensity physical activity (LPA; minutes with a step frequency ≤110 steps min$^{-1}$ in proportion to waking time)[40]. Waking time was determined according to Van der Berg et al.[56].

**Body composition**. Body composition (fat and fat-free mass) was determined at 8:00 a.m. after an overnight fast from 10:00 p.m. the previous evening using air displacement plethysmography (BodPod®, COSMED, Inc., Rome, Italy).

**Muscle strength**. Muscle contractile performance was measured using the Biodex System 3 Pro dynamometer (Biodex® Medical Systems, Inc., Shirley, NY, USA). For the measurements, the participants were stabilized in the device with shoulder, leg, and abdominal straps to prevent compensatory movement. The test was performed with the left leg in all participants. To measure maximal muscle strength, each participant performed 30 consecutive knee extension and flexion movements (range of motion 120 degrees s$^{-1}$). The peak torque of each extension and flexion was recorded and maximal isokinetic knee-extensor and knee-flexor torque was defined of the highest peak torque and corrected for fat-free mass (Nm kgFFM$^{-1}$).

**Resting energy expenditure**. Resting energy expenditure (REE) and substrate utilization were measured at 8:00 a.m., after an overnight fast from 10:00 p.m. the preceding evening. Gas exchange was measured by open-circuit respirometry with an automated ventilated hood system for 45 min. During the measurement, participants lay on a bed in a supine position. Data from the first 5 and last 5 minutes were omitted. REE data was adjusted for fat-free mass by calculating 'REE residuals', essentially according to Ravussin et al.[57].

**Submaximal exercise test and exercise energy expenditure**. After assessing resting energy expenditure, participants performed a 1-h submaximal exercise bout in the fasted state on an electronically braked cycle ergometer. To reach equal levels of exercise intensity, submaximal cycle test was performed at 50% of W$_\mathrm{max}$ as measured during the maximal aerobic cycling test. Participants were instructed to pedal at a controlled cadence between 60 and 70 revolutions per minute. To calculate exercise energy expenditure (EEE) and substrate oxidation, O₂ consumption and CO₂ production were recorded using indirect calorimetry for 15 min at two time points ($t = 15\,\mathrm{min}$ and $t = 45\,\mathrm{min}$). The submaximal cycle test was performed at, on average, 109[27] Watt versus 74[58] Watt in young versus older adults, respectively ($p < 0.001$). The TO individuals performed the submaximal cycle test at higher absolute power (on average 97[26] Watt) in comparison with O (74[58] Watt, $p = 0.030$) and IO (60[24] Watt, $p = 0.017$).

**Calculations of energy expenditure and exercise efficiency**. Energy expenditure was measured in rest, upon insulin stimulation during the hyperinsulinemic-euglycemic clamp, and during the submaximal cycle test using indirect calorimetry. The Weir equation[59] was used to calculate whole-body energy expenditure from measurements of $O_2$ consumption and $CO_2$ production. Carbohydrate and fat oxidation rates were calculated using the non-protein equations by Péronnet and Massicotte[54].

During submaximal exercise test, gross energy efficiency (GEE) was computed as the power output (watts converted to kJ min$^{-1}$) over exercise energy expenditure (in kJ min$^{-1}$) during the 1-h submaximal bike test and expressed as a ratio (Eq. 1) as described previously[60]. Mean values of work, $VO_2$, $VCO_2$ were averaged over 15-min periods at two time points ($t = 15$ min and $t = 45$ min). Data from the first 5 min and last 2 min were omitted.

$$GEE\ (\%) = \left( \frac{Work\ (kJ\,min^{-1})}{EEE\ (kJ\,min^{-1})} \right) . 100 \qquad (1)$$

Net energy efficiency (NEE) was measured from the submaximal test as power output (watts converted to kJ min$^{-1}$) over EE during exercise (EEE) minus resting EE (REE) (Eq. 2) as described by[60]. REE was measured on the same day as described above.

$$NEE\ (\%) = \left( \frac{Work\ (kJ\,min^{-1})}{EEE\ (kJ\,min^{-1}) - REE\ (kJ\,min^{-1})} \right) . 100 \qquad (2)$$

**Skeletal muscle biopsy**. Prior to the submaximal exercise test, a muscle biopsy was taken from the *m. vastus lateralis* under local anesthesia (1.0% lidocaine without epinephrine) according to the Bergström method[61]. Part of the biopsy was immediately placed in an ice-cold preservation medium (BIOPS, OROBOROS Instruments, Innsbruck, Austria) and used to measure ex vivo mitochondrial oxidative capacity. The remaining part of the muscle biopsy was immediately frozen in melting isopentane and stored at −80 °C until further analysis.

**Western blot analysis**. Mitochondrial content was assessed by mitochondrial OXPHOS protein expression using western blot analyses in Bioplex-lysates of human muscle tissue as previously described[62]. Equal amounts of protein were loaded 4–12% Bolt gradient gels (Novex, Thermo Fisher Scientific, Bleiswijk, The Netherlands). Proteins were transferred to nitrocellulose with the Trans-Blot Turbo transfer system (Bio-Rad Laboratories). Primary antibodies contained a cocktail of mouse monoclonal antibodies directed against human OXPHOS (dilution 1:10,000; ab110411, Abcam, Cambridge, UK). The hOxPHOS proteins were detected using secondary antibodies conjugated with IRDye680 or IRDye800 and were quantified with the CLx Odyssey Near-Infrared Imager (Li-COR, Westburg, Leusden, The Netherlands).

**Ex vivo high-resolution respirometry**. Permeabilized skeletal muscle fibers were immediately prepared from the muscle tissue collected in the preservation medium, as described previously[63]. Subsequently, the permeabilized muscle fibers (~2.5 mg wet weight) were analyzed for mitochondrial function using an oxygraph (OROBOROS Instruments, Innsbruck, Austria), according to Hoeks et al.[63]. To prevent oxygen limitation, the respiration chambers were hyper-oxygenated up to ~400 µmol L$^{-1}$ $O_2$. Subsequently, two different multi-substrate/inhibition protocols were used in which substrates and inhibitors were added consecutively at saturating concentrations. State 2 respiration was measured after the addition of malate (4 mmol L$^{-1}$) plus octanoyl-carnitine (50 µmol L$^{-1}$) or malate (4 mmol L$^{-1}$) plus glutamate (10 mmol L$^{-1}$). Subsequently, an excess of 2 mmol L$^{-1}$ of ADP was added to determine coupled (state 3) respiration. Coupled respiration was then maximized with convergent electron input through Complex I and Complex II by adding succinate (10 mmol L$^{-1}$). Finally, the chemical uncoupler carbonylcyanide-4-(trifluoromethoxy)-phenylhydrazone (FCCP) was titrated to assess the maximal capacity of the electron transport chain (state 3u respiration) or oligomycin (2 µg ml$^{-1}$) was added to assess the respiration not coupled to ATP synthesis (state 4o respiration). The integrity of the outer mitochondrial membrane was assessed by the addition of cytochrome C (10 µmol L$^{-1}$) upon maximal coupled respiration. If cytochrome C increased oxygen consumption >15%, the measurement was excluded to assure the viability and quality of the muscle mitochondrial measurement. All measurements were performed in quadruplicate.

**Magnetic resonance spectroscopy (muscle volume and PCr recovery)**. All magnetic resonance (MR) experiments were performed on the same day on a 3T whole-body MRI scanner (Achieva 3T-X; Philips Healthcare, Best, The Netherlands). To standardize food intake, participants consumed a light lunch at noon and remained fasted until completion of all MR experiments. Participants reported to the university at 2:30 p.m. and were seated in the waiting room for at least 30 min to minimize the possible effect of prior muscle activity. At 3:00 p.m., participants were positioned supine in the MR scanner to determine muscle volume with a series of T1-weighted images of the upper leg (slice thickness = 10.0 mm, no gap between slices, in-plane resolution = 0.78 × 0.78 mm). A custom-written MATLAB 2016a script (The Mathworks Inc. Natick, MA, USA) was used to semi-automatically segment adipose tissue and muscle and quantify muscle volume. The muscle segmentation was performed in the consecutive slices between the lower

end of the *m. rectus femoris* and the lower end of the *m. gluteus maximus*. Subsequently, phosphorus magnetic resonance spectroscopy ($^{31}$P-MRS) was performed to measure in vivo mitochondrial function in *m. vastus lateralis* as previously described[64], using a 6 cm surface coil. A series of 150 unlocalized $^{31}$P-spectra was acquired using the following parameters: single acquisitions (NSA = 1); repetition time (TR) = 4000 ms; spectral bandwidth = 3000 Hz; number of points = 1024. The 150 spectra were divided into 10 spectra at rest, 70 spectra during knee-extension exercise, and 70 spectra during recovery. Exercise within the scanner was performed to an auditory cue (0.5 Hz) in a custom-built knee-extension device with adjustable weight. The intensity was chosen to correspond to 50–60% of the predetermined maximum weight (determined on a separate day). Spectra were analyzed with a custom-made MATLAB 2016a script. PCr, ATP, and inorganic phosphate peaks were fitted, and pH was determined. The PCr recovery was fitted with a mono-exponential function and the rate constant (κ in s$^{-1}$) was determined as previously described[64]. The rate constant κ of PCr resynthesis is almost entirely dependent on ATP produced by oxidative phosphorylation and can therefore be used as a parameter of in vivo oxidative capacity[65].

**Walking performance and unperturbed and perturbed walking stability**. To further determine physical function, walking performance and stability were quantitatively assessed during a self-paced 6MWT, during multiple fixed-speed gait trials, and during a repeated balance disturbance trial. All tasks were measured using a dual-belt, force plate-instrumented (1000 Hz) treadmill within a virtual environment (Computer Assisted Rehabilitation Environment Extended, CAREN; Motekforce Link, Amsterdam, The Netherlands) and a 12-camera motion capture system (100 Hz; Vicon Motion Systems, Oxford, UK).

The 6MWT began with an explanation and one familiarization session (including two "start-ups" to let participants become accustomed to the self-paced function) on the CAREN prior to completing the recorded 6MWT. The 6-minute walk distance was taken to represent performance[66]. To assess gait variability, the means and coefficients of variation of step time, step length, step width, and double support time were measured during fixed-speed trials. This session again began with familiarization trials, followed by recorded trials (2–3 min – minimum of 120 steps) at speed of 0.4–1.8 m s$^{-1}$ in 0.2 m s$^{-1}$ steps[67]. These trials were then analyzed to calculate a stability-normalized walking speed for each participant individually for use in the gait perturbation trial, set for a margin of stability (MoS – see below; Hof et al.[25]) of 0.05 m as described previously[68], ensuring a comparable baseline gait stability for all groups[69].

The perturbation trial included 3–4 min of unperturbed walking, followed by 10 unannounced unilateral treadmill belt acceleration perturbations (every 30–90 s) as reported previously[36]. The first perturbations to each limb were analyzed (Pert1$_R$ and Pert2$_L$), representing novel disturbances, as well as the ninth perturbation (final left leg perturbation; Pert9$_L$) to indicate adaptation in gait following eight repeated perturbations. The anteroposterior MoS at foot touchdown were calculated[25], adapted for our validated reduced kinematic model[70]. The MoS was calculated for the following steps: the mean MoS of the eleventh to second last step before each perturbation (Base); the final step before each perturbation (Pre); and the first eight recovery steps following each perturbation (Post1–8).

The number of steps needed to return to pre-perturbation stability was determined by calculating the number of steps within 0.05 m of the MoS value of Base for each individual, counting back from the eighth recovery step, using custom-written R code (R version 3.6.0).

For all three gait tasks, initial data processing was conducted in MATLAB 2016a. The three-dimensional coordinates of the markers were filtered using a low pass second-order Butterworth filter (zero-phase) with a 12 Hz cut-off frequency. Foot touchdown and toe-off were detected using a combined method of force plate data (50 N threshold) and foot marker data, as described in detail previously[67]. Only three individuals in the IO group were able to complete the perturbation trial; as such the IO group was excluded from the perturbation analyses.

**Statistics**. Statistical analyses were performed using SPSS 26.0 (IBM, Chicago, IL, USA). Data are reported as mean (SD) unless stated otherwise. Analyses were performed for $n = 59$ unless specified otherwise in the table and the figure descriptions. The distribution of sex across groups was determined by $\chi^2$ test. In relation to the first research aim, differences between Y and O were tested by two-sided, independent samples *t*-test. To address the second research aim, group differences between O, TO, and IO were determined by means of a one-way analysis of variance (ANOVA) with Tukey's post-hoc test or by a two-sided Kruskal–Wallis test with Bonferroni correction, as appropriate. Of note: the same group of older adults with normal physical activity (O) was included in both comparisons.

GraphPad Prism version 8.02 for Windows was used for statistical analysis of the gait tasks (GraphPad Software Inc., La Jolla, California, USA). For the fixed-speed gait trials, mixed-effects models using the restricted maximum likelihood method with group (Y and O or O and TO) and walking speed (repeated measure: 0.4, 0.6, 0.8, 1.0, 1.2, 1.4, and 1.6 m s$^{-1}$) as factors were conducted for the mean gait parameters and their variability. For the perturbation trials, two-way repeated measures ANOVAs with group (Y and O or O and TO) and step (repeated measures; Base, Pre, Post1–8) as factors were conducted for perturbations Pert1$_R$, Pert2$_L$, and Pert9$_L$ with Dunnett's or Sidak's tests for multiple comparisons.

Regarding the number of recovery steps required, Wilcoxon matched-pairs signed rank tests were used to test for differences between the perturbations with the groups and Mann–Whitney tests were used to compare the groups to each other.

To assess associations between variables, we conducted bivariate Pearson or Spearman correlation and partial correlation analyses corrected for sex, age, and BMI. For all tests $p < 0.05$ was considered statistically significant.

**Reporting summary**. Further information on research design is available in the Nature Research Reporting Summary linked to this article.

## Data availability
Source data for all figures and tables have been provided with this paper (Source Data). All other data supporting the findings of this study are available from the corresponding author upon reasonable request.

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

## Acknowledgements

The project is organized by and executed under the auspices of TiFN, a public-private partnership on precompetitive research in food and nutrition. Funding for this research was obtained from Danone Nutricia Research, Friesland Campina, the Netherlands Organisation for Scientific Research, and the Top-sector Agri&Food.

## Author contributions

L.G. designed and performed the experiments, analyzed the data, and wrote the manuscript. N.J.C., C.M., C.E.F., L.B., Y.M.H.B., J.M., J.A.J., E.M-K., G.S., and B.H., assisted during the experiments, the data analysis, and reviewed and edited the manuscript. V.B.S., J.V.d.B., M.C.E.B., K.M., P.S., and J.H. contributed to the design of the study, interpretation of the data, and reviewed and edited the manuscript.

## Competing interests

J.V.d.B and M.C.E.B are affiliated with Danone Nutricia Research and Friesland-Campina, respectively. Friesland-Campina and Danone Nutricia Research are sponsors of the TiFN program and partly financed the project. The other authors declare no competing interests.
