## [Peer Review File · Nature Communications]

Reviewer comments, first round -

Reviewer #1 (Remarks to the Author):

The manuscript by Grevendonk et al., describes the results of a clinical study consisting of two cross sectional comparisons between, 1. young and old (similar PA levels) groups, and 2. Old trained, old normal PA and old physically impaired. All study participants were well phenotyped over 5 study visits focusing on physical function, muscle function and mitochondrial energetics (both in vivo and ex vivo from muscle biopsies). The goal of the study was to assess the impact of physical activity and exercise training on apparent age-associated decline in mitochondrial function. The principal finding is that aging is associated with a decline in mitochondrial energetics, exercise efficiency, muscle function, and insulin sensitivity among other variables – despite – maintenance of PA levels above the recommended guidelines for PA. This finding is in-line with some but not all (PMID: 24371120, PMID: 17376148, PMID: 22192354) comparisons of young and old while controlling for fitness/PA. Indeed, the data will be an important contribution to the literature and emphasizes the point that further studies are needed to understand how variation in PA impact the effect of aging on skeletal muscle mitochondrial function. The manuscript is well written, and the studies appear to be well conducted by a group with extensive experience and track record with clinical studies of metabolism and muscle mitochondria. The findings are interpreted in an appropriate manner and all relevant literature is considered and cited. That said, the principal findings are confirmatory for the most part, rather than novel and impactful, as might be expected for publication in Nature Communications. In addition, this reviewer identified a number of serious concerns (as outlined below) which together lessen enthusiasm for this manuscript.

Other Concerns:

The presentation of the results is not conventional. Is the old normal PA group in the Y v O and the O v TO v IO comparisons the same group? If so, this is not explicitly stated in the methods and could be perceived as misleading. Why not just compare all 4 groups in one comparison? This would simplify the presentation of results.

The IO group is described as both physically impaired and pre-frail. How was pre-frailty defined? This seems to be missing from the methods section.

Differences in group adiposity is a likely a confounder for some of the assessments. The O and IO groups are both overweight. In this case, normalizing variables to BW (Biodex peak torque data) will result in confounding due to increased adiposity in the IO and O group. Suggest normalizing to FFM?

Insulin sensitivity is expressed as the M-value (mg/kg/min). Is this normalized to Kg body weight or FFM? Again, if body weight this could present a problem due to group differences in adiposity.

Glucose tracer was used during the clamps. To what degree was HGP suppressed during insulin stimulated steady state? Also, was plasma insulin similar between groups during steady state? If not, do the data need to be normalized to plasma insulin?

The resting energy expenditure data is presented as Kj/min. Suggest allometric modeling to normalize REE for quantitative differences in body weight and/or body composition.

The lack of a difference in PCR recovery between OT, and the O/IO groups is difficult to explain.

Reviewer #2 (Remarks to the Author):

In this manuscript, Grevendonk and co-workers elegantly discuss the association of skeletal

muscle mitochondrial capacity to exercise capacity, efficiency, gait ability, muscle function, and insulin sensitivity related to aging. They report a link between muscle mitochondrial function and age-associated deterioration of skeletal muscle. This study brings about an insightful association between the ATP generating organelle and many body functions and attribute age-related decline to mainly to mitochondrial decline as measured in skeletal muscle.

Although this reviewer has some modest disagreements on some aspect of the conclusions and some questions and suggestions, believe that this is an important paper with substantial amount of carefully collected human data.

Specific points:

The main strength are the analytical rigor and methodologies especially measuring mitochondrial functions based on both ex vivo and in vivo approaches. It may help to demonstrate a correlation between the two approaches. The different activity levels are carefully documented in young and old. The investigators may expand their analysis/discussion to consider other factors such as cardiac output and tissue perfusion that may determine VO₂ max besides enhanced mitochondrial respiration and discuss some of the limitations of the study as well as be more open minded on the final conclusions.

1. It is important to recognize that muscle mitochondrial ATP production (Ref #75) is not different in vigorously trained older people from sedentary young people. It also has been shown in publications that while modest aerobic activity levels as in combined exercise training fails to normalize muscle mitochondrial respiration and insulin sensitivity rigorous exercise programs such as high-intensity interval training can normalize muscle mitochondrial respiration (Robinson MM et al Cell Metabolism 2017). The above and many other studies also show the importance of intensity of exercise as a determinant of ameliorating any age-related biological functions.

2. Of interest, there is a lack of close association between increment of skeletal muscle mitochondrial respiration following aerobic exercise and VO₂ max as shown in Robinson paper suggesting that VO₂ max depends also on cardiac output and tissue perfusion. Cardiac changes occurring with age rarely completely reverse on aerobic training unlike muscle mitochondrial respiration.

3. The authors may consider if there is any correlation between differences in VO₂ max and muscle mitochondrial respiration (maximum respiration-state 3). This association seems to be less obvious as people become older based on previous studies.

4. There are many publications suggesting that the association between mitochondrial respiration and insulin sensitivity is rather weak and has no evidence of cost-effect relationship. Aerobic exercise training improves both mitochondrial respiration and insulin sensitivity while resistance exercise has only minimal effect on muscle mitochondrial respiration although robustly improves insulin sensitivity. Also as shown by Irving BA et al 2011 that although high intensity exercise training was equally effective in offspring of mothers with T2D and healthy controls in enhancing muscle mitochondrial respiration insulin sensitivity did not improve in offspring of T2D.

5. Adiposity especially in the abdominal regions and activity levels are key determinants of insulin sensitivity although no evidence of adiposity as a determinant of mitochondrial respiration. In the current study, older people in general have more adiposity that may be a key factor besides activity level contributing to decline in insulin sensitivity. There are studies in the literature showing that aerobic exercise program or those people who are regularly exercise have no age-related decline in insulin sensitivity. These factors need to be considered in discussion.

Reviewer #3 (Remarks to the Author):

This is a very thorough experimental study of an important topic: mitochondrial content and function in young and older subjects, and the effects of physical activity vs. aging per se. I admire the difficulty involved in completing the work with a sufficiently high number of subjects in each group, as well as the breadth of analyses involved. The work is technically very well done and the conclusions are sound.

Major comment;

1. The MS is padded somewhat with interesting, but not relevant information on gait, stability etc which have virtually nothing to do with mitochondria. Figures 4, 5, 11, 12 and 13 and table 3 are irrelevant to the discussion. They should be removed, or at the very least placed in the

supplementary information and referred to only peripherally. There is no relationship of these measured items to mitochondrial function at all. they seem displaced in this context and they are poorly annotated (x-axis) and difficult to understand as described.

Response to referees

Reviewer #1 (Remarks to the Author):

The manuscript by Grevendonk et al., describes the results of a clinical study consisting of two cross sectional comparisons between, 1. young and old (similar PA levels) groups, and 2. Old trained, old normal PA and old physically impaired. All study participants were well phenotyped over 5 study visits focusing on physical function, muscle function and mitochondrial energetics (both in vivo and ex vivo from muscle biopsies). The goal of the study was to assess the impact of physical activity and exercise training on apparent age-associated decline in mitochondrial function. The principal finding is that aging is associated with a decline in mitochondrial energetics, exercise efficiency, muscle function, and insulin sensitivity among other variables – despite – maintenance of PA levels above the recommended guidelines for PA. This finding is in-line with some but not all (PMID: 24371120, PMID: 17376148, PMID: 22192354) comparisons of young and old while controlling for fitness/PA. Indeed, the data will be an important contribution to the literature and emphasizes the point that further studies are needed to understand how variation in PA impact the effect of aging on skeletal muscle mitochondrial function. The manuscript is well written, and the studies appear to be well conducted by a group with extensive experience and track record with clinical studies of metabolism and muscle mitochondria. The findings are interpreted in an appropriate manner and all relevant literature is considered and cited. That said, the principal findings are confirmatory for the most part, rather than novel and impactful, as might be expected for publication in Nature Communications. In addition, this reviewer identified a number of serious concerns (as outlined below) which together lessen enthusiasm for this manuscript.

We thank the reviewer for the very constructive and helpful comments. We have made several changes to the manuscript in response to the comments provided (see below).

Major Concerns:

The presentation of the results is not conventional. Is the old normal PA group in the Y v O and the O v TO v IO comparisons the same group? If so, this is not explicitly stated in the methods and could be perceived as misleading. Why not just compare all 4 groups in one comparison? This would simplify the presentation of results.

We have indeed specified four groups in this cross-sectional study; young individuals with normal physical activity (Y, 20–30 years), older adults with normal physical activity (O, 65–80 years), trained older adults (TO, 65–80 years) and physically impaired older adults (IO, 65–80 years). As explained in the introduction of the manuscript, the aim of the study was twofold: 1) to assess if mitochondrial function is reduced in older compared to young participants with a similar level of habitual PA and 2) to investigate the potential of regular exercise training in older adults for the maintenance of mitochondrial function. To address our first aim, we directly compared Y and O by t-test. Subsequently, a one-way ANOVA was performed to compare the three older groups in relation to the second research question. We explicitly decided not to compare all 4 groups in one statistical comparison since we felt that - given our research aims - it was irrelevant to make comparisons between Y and TO

and between Y and IO. This is also the reason that we display the results in separate figures. In this context, the older group with normal physical activity (O) indeed is the same group in both comparisons and we agree with the reviewer that this could have been expressed more clearly. Therefore, we now explicitly state this in the methods section of the manuscript.

The IO group is described as both physically impaired and pre-frail. How was pre-frailty defined? This seems to be missing from the methods section.

Thank you for pointing this out. This is an unfortunate mistake, as indeed, we cannot refer to the IO group as 'pre-frail' since this group is solely defined on the score obtained in the standard physical performance battery (SPPB<9). The qualification 'pre-frail' has - in relation to the IO group - been removed from the manuscript and replaced by 'physically impaired'.

Differences in group adiposity is a likely a confounder for some of the assessments. The O and IO groups are both overweight. In this case, normalizing variables to BW (Biodex peak torque data) will result in confounding due to increased adiposity in the IO and O group. Suggest normalizing to FFM?

We appreciate this suggestion and have now normalized the (Biodex) muscle strength data for fat-free mass (FFM). In essence this normalization did not affect the differences between groups, but the previously observed correlation between muscle strength and mitochondrial function was no longer present. The figures, tables and manuscript text are amended accordingly.

Insulin sensitivity is expressed as the M-value (mg/kg/min). Is this normalized to Kg body weight or FFM? Again, if body weight this could present a problem due to group differences in adiposity.

Glucose tracer was used during the clamps. To what degree was HGP suppressed during insulin stimulated steady state? Also, was plasma insulin similar between groups during steady state? If not, do the data need to be normalized to plasma insulin?

The M-value data was indeed normalized to body mass, which is most common for M-value. However, M-value gives limited information on the more detailed aspects of insulin sensitivity. Therefore, we have now analyzed the plasma samples from the clamp to determine the glucose tracer kinetics and included new data on suppression of hepatic glucose output and peripheral glucose uptake (Rd) in the manuscript. In addition, we also measured plasma insulin levels during the steady-state of the clamp to correct for potential age-related alterations in insulin levels during the clamp. Indeed, young vs. old adults displayed differences in insulin levels during the clamp, and therefore we calculated the insulin sensitivity index (S_i) from the Rd (from baseline to clamp steady-state) corrected for plasma insulin and glucose levels during the clamp, as described in Remie et al. Nat Comm (2021). Since we acknowledge that differences in adiposity could contribute to differences in insulin sensitivity and since insulin-stimulated peripheral glucose uptake is primarily accounted for by skeletal muscle, we also decided to present the S_i normalized for fat-free mass (FFM) in figure 2b and 5b. These new data show that S_i was lower in old vs. young adults ($p=0.014$) when expressed per kg body weight (data given in manuscript text), a difference that was largely retained when S_i was normalized for FFM ($p=0.068$, fig. 2b).

Trained old individuals tended ($p=0.072$) to have a higher S_i in comparison to old adults with normal physical activity, when expressed per kg body weight (data given in manuscript text). This difference was not observed ($p=0.165$) when S_i was normalized for FFM (fig. 5b). Insulin-induced suppression of EGP was similar in young vs. old, but also in old vs trained old adults. The abovementioned new data and changes have now been incorporated into the manuscript.

The resting energy expenditure data is presented as KJ/min. Suggest allometric modeling to normalize REE for quantitative differences in body weight and/or body composition.

Thank you for this suggestion. In response to this comment, we now adjusted the resting energy expenditure (REE) data for lean body mass, in essence according to Ravussin et al. (J Gerontol A Biol Sci Med Sci, 2015 Sep;70(9):1097-104). Thus, we calculated 'REE residuals', i.e. the difference between an individual's REE measured by indirect calorimetry and the REE predicted from a regression of REE as a function of fat-free mass (FFM) in all participants of the study.

The lack of a difference in PCR recovery between OT, and the O/IO groups is difficult to explain.

Based on the differences observed in oxygen consumption measurements in permeabilized muscle fibers, we were indeed also surprised to see that the PCr recovery rate was not statistically different between the older groups. We can only speculate that the PCr recovery measurement may be confounded by limiting systemic factors such as adequate perfusion and oxygen delivery during and after exercise, and other characteristics of the microenvironment, as mentioned in the manuscript discussion.

Reviewer #2 (Remarks to the Author):

In this manuscript, Grevendonk and co-workers elegantly discuss the association of skeletal muscle mitochondrial capacity to exercise capacity, efficiency, gait ability, muscle function, and insulin sensitivity related to aging. They report a link between muscle mitochondrial function and age-associated deterioration of skeletal muscle. This study brings about an insightful association between the ATP generating organelle and many body functions and attribute age-related decline to mainly to mitochondrial decline as measured in skeletal muscle.

Although this reviewer has some modest disagreements on some aspect of the conclusions and some questions and suggestions, believe that this is an important paper with substantial amount of carefully collected human dat.

Specific points:

The main strength are the analytical rigor and methodologies especially measuring mitochondrial functions based on both ex vivo and in vivo approaches. It may help to demonstrate a correlation between the two approaches. The different activity levels are carefully documented in young and old. The investigators may expand their analysis/discusion to consider other factors such as cardiac output and tissue perfusion that may determine VO₂ max besides enhanced mitochondrial respiration and discuss some of

the limitations of the study as well as be more open minded on the final conclusions.

Thank you for the positive evaluation of our manuscript. We now expanded our discussion, as outlined in the responses to the specific points below.

It is important to recognize that muscle mitochondrial ATP production (Ref #75) is not different in vigorously trained older people from sedentary young people. It also has been shown in publications that while modest aerobic activity levels as in combined exercise training fails to normalize muscle mitochondrial respiration and insulin sensitivity rigorous exercise programs such as high-intensity interval training can normalize muscle mitochondrial respiration (Robinson MM et al Cell Metabolism 2017). The above and many other studies also show the importance of intensity of exercise as a determinant of ameliorating any age-related biological functions.

Thank you for pointing out the relevance of exercise intensity in improving (age-related deteriorations in) mitochondrial function and muscle health; we have now incorporated this aspect into the discussion, and also cite the reference mentioned by the reviewer.

Of interest, there is a lack of close association between increment of skeletal muscle mitochondrial respiration following aerobic exercise and VO₂ max as shown in Robinson paper suggesting that VO₂ max depends also on cardiac output and tissue perfusion. Cardiac changes occurring with age rarely completely reverse on aerobic training unlike muscle mitochondrial respiration.

The authors may consider if there is any correlation between differences in VO₂ max and muscle mitochondrial respiration (maximum respiration-state 3). This association seems to be less obvious as people become older based on previous studies.

The reviewer may have missed that we did observe a correlation between VO₂max and maximally coupled (state 3) respiration, as listed in table 3 of the manuscript. A scatterplot for this correlation can be found below and is now also added to the manuscript (figure 7a).

Figure 1: correlation between maximally-coupled (state 3) respiration (MOGS3) and VO₂max in the entire cohort of young individuals with normal physical activity (Y), older adults with normal physical activity (O), trained older adults (TO) and physically impaired older adults (IO).

This correlation persisted if only the older groups were considered, indicating that - in our study - a relation between VO₂max and mitochondrial respiration was also observed at older age.

Figure 2: correlation between maximally-coupled (state 3) respiration (MOGS3) and VO₂max in the older adults with normal physical activity (O), trained older adults (TO) and physically impaired older adults (IO) only.

There are many publications suggesting that the association between mitochondrial respiration and insulin sensitivity is rather weak and has no evidence of cost-effect relationship. Aerobic exercise training improves both mitochondrial respiration and insulin sensitivity while resistance exercise has only minimal effect on muscle mitochondrial respiration although robustly improves insulin sensitivity. Also as shown by Irving BA et al 2011 that although high intensity exercise training was equally effective in offspring of mothers with T2D and healthy controls in enhancing muscle mitochondrial respiration insulin sensitivity did not improve in offspring of T2D.

Despite the fact that we observe a significant correlation between mitochondrial respiration and insulin sensitivity in the present study, we fully agree with the reviewer this is merely an association and cannot be interpreted as a cause and effect relation. We acknowledge that improvements in insulin sensitivity can also occur without changes in mitochondrial respiratory capacity and vice versa, and now mention this in the manuscript discussion.

5. Adiposity especially in the abdominal regions and activity levels are key determinants of insulin sensitivity although no evidence of adiposity as a determinant of mitochondrial respiration. In the current study, older people in general have more adiposity that may be a key factor besides activity level contributing to decline in insulin sensitivity. There are studies in the literature showing that aerobic exercise program or those people who are regularly exercise have no age-related decline in insulin sensitivity. These factors need to be considered in discussion.

We agree with the reviewer that differences in adiposity could contribute to the observed differences in insulin sensitivity. Therefore, we amended the discussion and now state that "Despite similar levels of habitual of PA in the current study, body fat was higher and insulin sensitivity was lower in the older participants when compared to young. In comparison with older adults with normal PA, trained older adults displayed higher levels of insulin sensitivity,

underscoring the ability of exercise training to improve insulin sensitivity and glucose uptake, also at an older age. However, since trained older adults also featured a lower adiposity, changes in body composition may (partly) mediate the observed differences in insulin sensitivity, independent of mitochondrial function"

Reviewer #3 (Remarks to the Author):

This is a very thorough experimental study of an important topic: mitochondrial content and function in young and older subjects, and the effects of physical activity vs. aging per se. I admire the difficulty involved in completing the work with a sufficiently high number of subjects in each group, as well as the breadth of analyses involved. The work is technically very well done and the conclusions are sound.

Thank you, we highly appreciate the positive assessment of this challenging study.

Major comment;

1. The MS is padded somewhat with interesting, but not relevant information on gait, stability etc which have virtually nothing to do with mitochondria. Figures 4, 5, 11, 12 and 13 and table 3 are irrelevant to the discussion. They should be removed, or at the very least placed in the supplementary information and referred to only peripherally. There is no relationship of these measured items to mitochondrial function at all. They seem displaced in this context and they are poorly annotated (x-axis) and difficult to understand as described.

The assessment of gait stability and adaptability was performed as a measure for muscle coordination and balance, and deemed relevant for the older population in relation to falls risk and physical independence. We do agree with the reviewer though that the amount of information on this topic is somewhat overwhelming. Therefore, we decided to substantially shorten the manuscript text related to the gait outcomes and to move the majority of text and details, including all figures, to the supplementary material.

Reviewer comments, second round –

Reviewer #1 (Remarks to the Author):

The authors have adequately addressed the concerns that I raised regarding presentation of glucose clamp and indirect calorimetry data (among others).

Reviewer #2 (Remarks to the Author):

The authors have answered most of my concerns adequately